# Ab initio machine-learning unveils strong anharmonicity in non-Arrhenius self-diffusion of tungsten

Xi Zhang ®[1] ✉, Sergiy V. Divinski ®[2] & Blazej Grabowski ®[1]

The knowledge of diffusion mechanisms in materials is crucial for predicting their high-temperature performance and stability, yet accurately capturing the underlying physics like thermal effects remains challenging. In particular, the origin of the experimentally observed non-Arrhenius diffusion behavior has remained elusive, largely due to the lack of effective computational tools. Here we propose an efficient ab initio framework to compute the Gibbs energy of the transition state in vacancy-mediated diffusion including the relevant thermal excitations at the density-functional-theory level. With the aid of a bespoke machine-learning interatomic potential, the temperature-dependent vacancy formation and migration Gibbs energies of the prototype system body-centered cubic (BCC) tungsten are shown to be strongly affected by anharmonicity. This finding explains the physical origin of the experimentally observed non-Arrhenius behavior of tungsten self-diffusion. A remarkable agreement between the calculated and experimental temperature-dependent self-diffusivity and, in particular, its curvature is revealed. The proposed computational framework is robust and broadly applicable, as evidenced by first tests for a hexagonal close-packed (HCP) multicomponent high-entropy alloy. The successful applications underscore the attainability of an accurate ab initio diffusion database.

Understanding atomic diffusion is of fundamental importance in developing materials with controlled mechanical and functional properties. For the majority of metals, the temperature dependence of thermally activated, vacancy-mediated diffusion is well assessed from countless studies. The common knowledge expects a linear Arrhenius behavior. Magnetic[1,2], chemical order-disorder[3,4], or structural[5] transformations might induce characteristic deviations from the Arrhenius-type temperature dependence. When phase transformations do not interfere, the logarithm of the diffusion rate, $D$, is assumed to scale linearly with the inverse temperature, $T^{-1}$,

$$\ln D = -(Q/k_B)T^{-1} + \ln D_0, \qquad (1)$$

with the slope, $-Q/k_B$, and the intercept, $\ln D_0$, controlled by the activation energy $Q$ and the prefactor $D_0$ ($k_B$: Boltzmann constant). Both $Q$ and $D_0$ are generally assumed to be *independent* or very weakly dependent on temperature[6].

The universality of this widespread assumption is, however, limited, especially close to the melting point[6]. Diffusion measurements covering *wide* temperature intervals revealed deviations from linearity[7–12]. Textbook knowledge explains this "anomaly" of substitutional diffusion with di-vacancies in addition to mono-vacancies[6,13,14]. Various measurements[15–17] were interpreted as supporting the di-vacancy explanation, which grew into an undisputed paradigm. Nevertheless, some researchers proposed an alternative explanation in terms of temperature-dependent activation energies[18–22] indicating

[1]Institute for Materials Science, University of Stuttgart, D-70569 Stuttgart, Germany. [2]Institute of Materials Physics, University of Münster, 48149 Münster, Germany. ✉e-mail: xi.zhang@imw.uni-stuttgart.de

**Fig. 1 | TSTI workflow.** Schematic illustration of the proposed transition state thermodynamic integration (TSTI) approach (NEB = nudged elastic band; TS = transition state; cfgs = configurations (sampled from MD); $T^{el}$ = electronic temperature).

problems with the mono-/di-vacancy interpretation even in simple metals. Temperature-dependent vacancy formation energies driven by anharmonic vibrations were indeed uncovered by recent ab initio simulations[23,24]. Temperature-dependent vacancy energies were also found for self-diffusion in Mo[25].

These theoretical findings call for a conceptual revision of the long-standing paradigm. However, a paradigm shift requires strong support. Therefore, not only further ab initio simulations of defect energetics are required, but in particular methodological advances to enable access to reliable predictions at high temperatures.

While efficient high-accuracy ab initio techniques are available for vacancy formation[23,26], state-of-the-art ab initio approaches approximate migration Gibbs energies by (quasi)harmonic transition state theory (hTST)[27], utilizing either the full quasiharmonic free energy[28] or a simplified Vineyard formula[27] with explicit phonon calculations[29]. It will be shown below that hTST is unrealistic at elevated temperatures and causes significant errors.

The most accurate approach to exploring the full vibrational space is thermodynamic integration. A crucial requirement for thermodynamic integration is "sufficient" phase stability, i.e., the target phase should be dynamically stable in the temperature range of interest. Certain correlated lattice instabilities, e.g., the hcp to dhcp transformation as observed for hcp Ni[30], are suppressed by increasing the size of the simulation cell. Other undesirable transformations may be inhibited by restricting the cell to smaller sizes[31]. For an intrinsically localized instability as experienced by a diffusing atom at the energetic saddle point, such measures will not be effective. Although finite-temperature path-based approaches such as the finite temperature string method[32] and the mean-force based integration[33,34] have been developed to include the full vibrational contribution to the activation free energy, their practical application in an ab initio framework is not straightforward.

Here, we propose an efficient and accurate ab initio method that overcomes the dynamical instability problem of the transition state. We show that with the introduced stabilization scheme—implemented in a machine-learning-assisted thermodynamic-integration + direct-upsampling framework—the full temperature dependence of vacancy migration Gibbs energies can be efficiently calculated, including all relevant thermal excitations with density-functional-theory (DFT) accuracy. Together with the vacancy formation Gibbs energies, likewise computable with the direct upsampling technique, accurate ab

initio self-diffusivities become available, even for high melting systems.

## Results

### Transition state thermodynamic integration

Figure 1 provides the general workflow of the introduced *transition state thermodynamic integration* (TSTI) approach. The key parts are highlighted by the gray-blue boxes. At the core lies the actual TSTI calculation from a stabilized dynamical matrix to a highly optimized machine-learning potential, specifically a moment tensor potential (MTP)[35]. The TSTI calculation is followed by direct upsampling to achieve DFT accuracy.

The MTP is trained on a large DFT dataset (2591 structures) including bulk, vacancy, and transition-state configurations sampled from high-temperature MD. The root-mean-square error (RMSE) is only 1.9 meV/atom in energies and 0.15 eV/Å in forces, respectively. Consequently, the MD energies (2591 energies in total) and forces (nearly one million forces in total) predicted by the MTP show a strong correlation with the DFT data, cf. Fig. 2. The good performance of the MTP ensures an accurate description of the vibrational phase space for the considered bulk, vacancy, and transition-state configurations and boosts up the direct upsampling convergence. More details about the MTP are given in the Methods section.

The anharmonic free energy is expressed as

$$F^{ah} = \Delta F_{MTP}^{qh \to full} + \Delta F_{DFT}^{up}, \tag{2}$$

where the first term $\Delta F_{MTP}^{qh \to full}$ is obtained from the TSTI calculation to the MTP and the second term $\Delta F_{DFT}^{up}$ from free energy perturbation theory, i.e., the direct upsampling, to DFT. Specifically,

$$\Delta F_{MTP}^{qh \to full} = \int_0^1 d\lambda \, \langle E_s^{full}(\{\mathbf{R}_I\}) - E^{qh}(\{\mathbf{R}_I\}; \underline{\underline{D}}_s) \rangle_\lambda \tag{3}$$

represents a thermodynamic integration along the coupling parameter $\lambda \in [0, 1]$, with the thermodynamic average $\langle \ldots \rangle_\lambda$ of the difference between the quasiharmonic energy $E^{qh}$ and the *stabilized* full vibrational MTP energy $E_s^{full}$ (discussed below) computed for different configurations of the atomic coordinates $\{\mathbf{R}_I\}$. For each *fixed* value of $\lambda$, the atomic movement in the MD simulation is driven by forces derived from a linear combination of quasiharmonic and MTP forces.

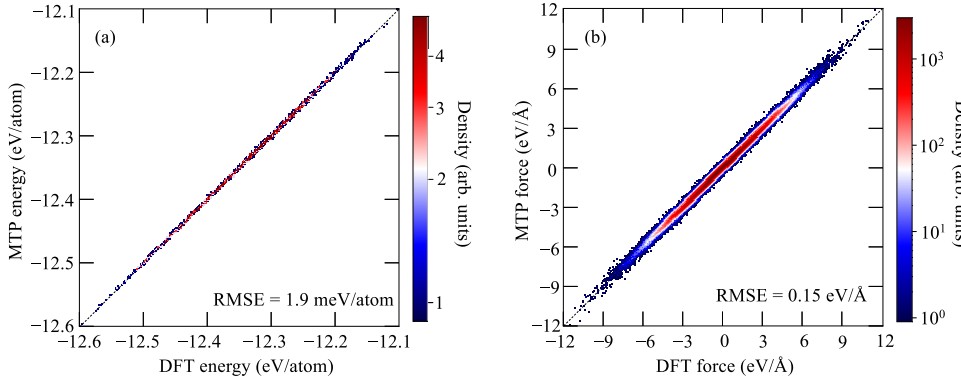

**Fig. 2 | Accuracy of the moment tensor potential (MTP).** Correlation plots of the energies (**a**) and forces (**b**) of the MTP vs. DFT and the corresponding root-mean-square errors (RMSEs).

A key ingredient to the here proposed approach is the *stabilized* dynamical matrix $\underline{\underline{D}}_s$ computed from the original, unstable dynamical matrix via

$$\underline{\underline{D}}_s = \underbrace{\underline{\mathbf{w}} \cdot \underline{\underline{\omega^2}} \cdot \underline{\mathbf{w}}^{-1}}_{3N-1\,\text{stable modes}} + \underbrace{\underline{\mathbf{w}} \cdot \underline{\omega_s}^2 \cdot \underline{\mathbf{w}}^{-1}}_{1\,\text{stabilized mode}}, \tag{4}$$

where

$$\underline{\underline{\omega^2}} = \mathrm{diag}(\omega_1^2, \ldots, \omega_{3N-1}^2, 0), \tag{5}$$

$$\underline{\omega_s}^2 = \mathrm{diag}(0, \ldots, 0, \alpha|\omega_{\text{imag}}^2|) \tag{6}$$

represent two $3N \times 3N$ diagonal matrices, which store the $3N - 1$ stable phonon frequencies $\omega_i$ and the single *stabilized* phonon frequency, respectively. The stabilization is achieved by taking the absolute value of the reaction coordinate frequency $\omega_{\text{imag}}^2$ times a tunable parameter $\alpha$. We set $\alpha$ to be 30, a value ensuring that the resulting stabilized frequency of the unstable mode, $\sqrt{\alpha|\omega_{\text{imag}}^2|}$, is higher than the highest stable frequency of the frequency spectrum of the transition state. More details and tests on the stabilization parameter $\alpha$ are provided in the Supplementary Information. Further in Eq. (4), $\underline{\mathbf{w}}$ is the eigenvector matrix of the original, unstable dynamical matrix.

The second crucial ingredient to our scheme is the stabilization of the MTP forces during the thermodynamic integration. This is achieved by replacing the forces along the unstable mode with the *stabilized* harmonic forces given by $\underline{\underline{D}}_s$. Specifically, for a given $\lambda$, the interatomic forces are obtained as

$$\mathcal{F}_\lambda = (1 - \lambda)\mathcal{F}_s^{\text{qh}} + \lambda\mathcal{F}_s^{\text{full}}, \tag{7}$$

with the forces $\mathcal{F}_s^{\text{qh}}$ corresponding to $\underline{\underline{D}}_s$ and with the *stabilized* MTP forces $\mathcal{F}_s^{\text{full}}$ computed by

$$\mathcal{F}_s^{\text{full}} = \mathcal{F}^{\text{MTP}}(\{\mathbf{R}_I\}) + [\mathcal{F}_s^{\text{qh}}(\{\mathbf{R}_I^{\text{sp}} + \mathbf{u}_I'\}) - \mathcal{F}^{\text{MTP}}(\{\mathbf{R}_I^{\text{sp}} + \mathbf{u}_I'\})], \tag{8}$$

where $\mathcal{F}^{\text{MTP}}$ are the original unstabilized MTP forces and

$$\mathbf{u}_I' = \left(\mathbf{u}_I \cdot \hat{\mathbf{w}}_{\text{imag}}\right) \cdot \hat{\mathbf{w}}_{\text{imag}}, \quad \mathbf{u}_I = \mathbf{R}_I - \mathbf{R}_I^{\text{sp}}, \tag{9}$$

with the saddle-point positions $\{\mathbf{R}_I^{\text{sp}}\}$ and the normalized eigenvector $\hat{\mathbf{w}}_{\text{imag}}$ of the unstable phonon mode. Consistently, the corresponding MTP potential energy entering Eq. (3) is computed as

$$E_s^{\text{full}}(\{\mathbf{R}_I\}) = E^{\text{MTP}}(\{\mathbf{R}_I\}) + [E^{\text{qh}}(\{\mathbf{R}_I^{\text{sp}} + \mathbf{u}_I'\}; \underline{\underline{D}}_s) - E^{\text{MTP}}(\{\mathbf{R}_I^{\text{sp}} + \mathbf{u}_I'\})], \tag{10}$$

where $E^{\text{MTP}}$ is the original unstabilized MTP potential energy.

The TSTI workflow relies upon and significantly advances the established computational framework for high-accuracy ab initio Gibbs energies for equilibrium states[36]. In particular, as highlighted by the two schematic plots in Fig. 1, the unstable mode (red dotted line), which drives the saddle-point configuration to the equilibrium, does not directly contribute to the vibrational free energy of the transition state. At *any* $\lambda$, the instability is replaced by the stable energy profile corresponding to $\alpha|\omega_{\text{imag}}^2|$ (red solid line). Since the stabilized energy profile along the thermodynamic-integration path remains unchanged, there is no explicit contribution of the reaction coordinate to the free energy difference in Eq. (2). (This is consistent with hTST in sampling the $(3N - 1)$ stable modes, albeit in the harmonic regime.) The indirect impact of the stabilization via the coupling to the stable modes is small and can be controlled with the $\alpha$ parameter (see Supplementary Information). The stabilization in TSTI is of crucial importance to restrict the vibrations to the vicinity of the saddle point, such as to enable the sampling of the anharmonicity (gray shaded area) from all other, $3N - 1$ stable modes (gray or black curves).

With TSTI, the anharmonicity of the transition state is captured at the accuracy level of the MTP. Upsampling based on molecular-dynamics snapshots with the full free-energy formula for $\Delta F_{\text{DFT}}^{\text{up}}$ ensures DFT accuracy. Electronic excitations and their coupling to atomic vibrations are computed in a second upsampling step utilizing finite temperature DFT[36,37]. Thus, the full set of free energy contributions can be computed for the transition state at the level of DFT accuracy.

## Temperature-dependent Gibbs energies of vacancy formation and migration

We use TSTI to calculate the Gibbs energy of vacancy migration $G_{\text{mig}}(T)$ for BCC W. The reference transition-state configuration, $\{\mathbf{R}_I^{\text{sp}}\}$, is obtained from the climbing-image nudged elastic band[38] method performed at zero Kelvin. At high temperatures, even close to the melting point, the MD migration trajectories suggest that, for symmetrical crystalline structures like BCC, the migration pathway remains the same (see Supplementary Fig. 6 and the related discussion). For complex molecular systems[39], migration pathways may be modified at high temperatures due to fewer symmetry restrictions, which would require a special analysis that is out of the scope of the present paper. We use the "standard" scheme of thermodynamic-integration + direct-upsampling[36] to calculate the Gibbs energy of vacancy formation $G_{\text{form}}(T)$. Previously, the temperature dependence of $G_{\text{form}}$ of BCC W was derived from fitting a single data point[40], and we now investigate the explicit temperature dependence on a dense temperature mesh. The self-diffusion

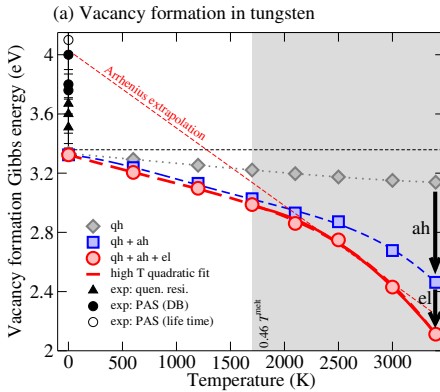

**Fig. 3 | Impact of anharmonicity and electrons on vacancy formation and migration in tungsten. a** Formation and **b** migration Gibbs energies from the proposed TSTI approach in comparison with experiments (black symbols with error bars showing the experimental uncertainty). Experimental techniques: **a** quenched-in resistivity[74–76], positron annihilation spectroscopy (PAS) Doppler broadening (DB)[77–79] and life time[78]; **b** recovery of quenched-in resistivity[75] and interrupted quenching[20]. The dotted/dashed lines connecting the computed data points (filled symbols) are fourth-order polynomial fits. The dashed red lines show the linear Arrhenius extrapolation of the Gibbs energies from the temperature range of 2600 K to 3000 K.

coefficient reads

$$D(T) = a_0^2(T) f_{BCC} \frac{k_B T}{h} \exp\left(-\frac{G_{form}(T) + G_{mig}(T)}{k_B T}\right), \quad (11)$$

where h is the Planck constant. The lattice constant at zero pressure $a_0(T)$ is obtained from the relative thermal expansion (DFT molecular dynamics) reported in ref. 41 and rescaled with the lattice constant at 0 K, 3.1723 Å, calculated here (see Supplementary Information for more details). The correlation factor for vacancy diffusion on the BCC lattice, $f_{BCC}$, takes the value of 0.7272[42]. For the DFT calculations, we employ the projector augmented wave (PAW) method[43] within the generalized gradient approximation (GGA) in the Perdew-Burke-Ernzerhof (PBE) parametrization[44] as implemented in VASP[45,46]. Further details are given in the Methods section.

Figure 3 presents the temperature-dependent Gibbs energies of vacancy formation and migration for BCC W. The red circles and lines correspond to the final Gibbs energies with all thermal contributions included. For both $G_{form}(T)$ and $G_{mig}(T)$, a strong and clearly *non-linear* temperature dependence is observed, particularly in the temperature range where diffusion is measured ($T > 0.46 T^{melt}$, gray shaded area). The computed high-temperature data are well reproduced by a quadratic fit (red solid lines), as observed before for formation Gibbs energies[23]. The energy reduction with temperature is particularly strong for vacancy formation and reaches −1.21 eV (−36%) at 3400 K as compared to the 0 K value.

The experimental data (black symbols) reflect extrapolations of high-temperature measurements utilizing the *linear* Arrhenius ansatz, i.e., $G_{form/mig} = H_{form/mig} − T S_{form/mig}$, with temperature-independent enthalpies ($H_{form/mig}$) and entropies ($S_{form/mig}$) of formation/migration. Although such a linear fitting ansatz is clearly inappropriate based on the present and previous results[23,24], we utilize it nevertheless to fit our exact data for the purpose of an unbiased comparison between theory and experiment. For both formation and migration, a linear Arrhenius fit of the full Gibbs energies between 2600 K and 3000 K shows a reasonable agreement with experiment, considering the significant scatter and uncertainty in the high-temperature measurements.

A comparison between the generally applied quasiharmonic approximation (gray diamonds) and the Gibbs energies featuring full vibrations (blue squares) unveils strong anharmonicity in the formation and migration of a vacancy. The Gibbs energies decrease significantly due to explicit anharmonicity as highlighted by the arrows labeled "ah"; max. −20% for the vacancy formation and −10% for the migration. The strong non-linear temperature dependence observed

for the full vibrational Gibbs energies is absent for the quasiharmonic approximation. Note that the Gibbs energy of migration is obtained from the difference of the Gibbs energy of the transition state and the vacancy supercell. Thus, the anharmonic contribution observed in Fig. 3b means that the transition state is even more anharmonic than the already strongly anharmonic vacancy supercell when compared to the bulk. This highlights that the so far well-established and generally applied quasiharmonic approximation is not adequate to describe the vibrational contribution in vacancy-mediated diffusion.

## Strong impact of electronic excitations

Electronic excitations have traditionally been considered less influential in state-of-the-art diffusion studies[29]. Thus in practice, explicit calculations of electronic free energy have often been omitted[29,47,48] or, when considered, relied on the ideal static lattice approximation[49].

The present computational framework enables us to explicitly account for electronic excitations and the coupling effect with thermal vibrations. As revealed in Fig. 3, electronic excitations including the coupling to vibrations provide a sizable contribution to the Gibbs energies (black arrows labeled with "el"). The corresponding decrease is −0.35 eV for the vacancy formation and −0.15 eV for the migration Gibbs energy.

The electronic impact can be understood from an analysis of the electronic density of states (eDOS). Figure 4 shows the electronic density of states at 3000 K extracted from molecular dynamics snapshots. The smoothening effect due to thermal vibrations (i.e., decrease of peaks and increase of valleys as the one close to the Fermi level) becomes stronger with the sequence of bulk to vacancy to transition state, as is observed from the mean eDOS's (solid lines in blue, red, black, respectively) each averaged over 120 MD snapshots (the lighter background with similar color tone). Despite the considerable variance in the eDOS's of the single snapshots, the standard errors of the mean values are about one order of magnitude smaller than the differences between the different types of structures (bulk, vacancy, and transition state) in particular at the valleys and peaks (see Supplementary Fig. 4), suggesting that the eDOS's are well converged. At high (electronic) temperatures, the Fermi function (dash-dotted curve shown in the right panel) re-populates the electrons from below the Fermi level to higher energy states above the Fermi level. The resulting holes and excited electrons are the origin of the electronic entropy contribution to the free energy. A *large* eDOS at the Fermi level facilitates many such excitations and thus implies a *largely negative* electronic free energy, as detailed in ref. 50. As observed around the valley region enlarged in the right panel of Fig. 4, the eDOS's close to the Fermi level increase

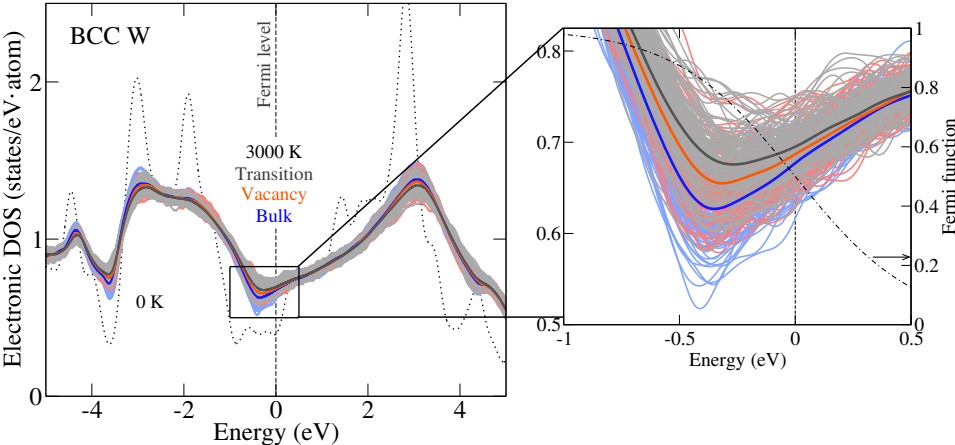

**Fig. 4 | Origin of the strong impact of electronic excitations.** The mean electronic densities of states of the bulk structure (blue), the vacancy structure (red), and the transition state structure (black) at 3000 K are shown. The corresponding DOS's of 120 MD snapshots from which the mean DOS is obtained are plotted in the background in a lighter version of the respective color. The right panel shows the enlarged view of the valley close to the Fermi level. The dash-dotted curve indicates the Fermi function at 3000 K.

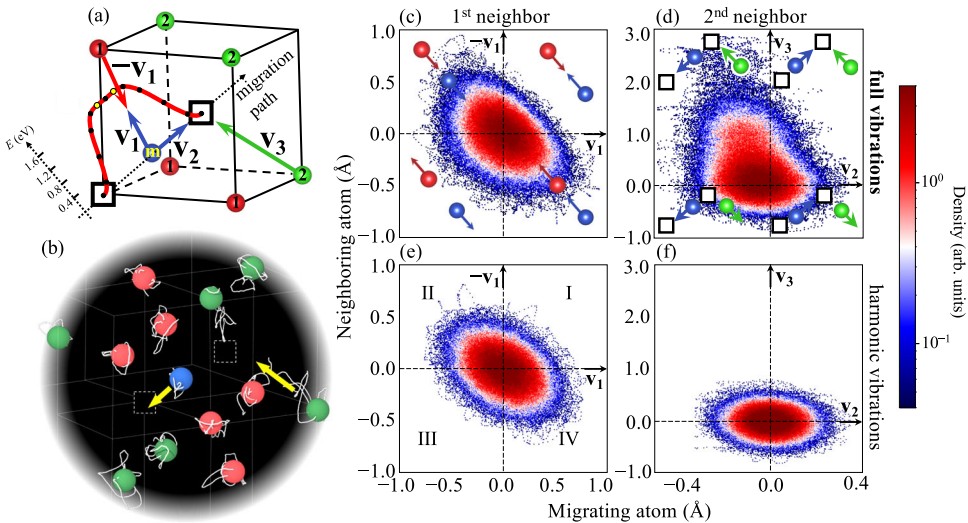

**Fig. 5 | Collective motion of the migrating atom and its neighborhood.**
**a** Migrating atom (blue atom labeled with "m") and its local environment (red atoms: 1st neighbors; green atoms: 2nd neighbors) in the transition state. The red energy profile shows the calculated NEB minimum energy path marked with the intermediate images (black dots) and the saddle points (yellow dots). **b** Real-time MD trajectory (the white line associated with each atom) of a short time frame of 300 MD steps (yellow arrows highlight the directions of the collective motion). **c–f** Distribution of the correlated vibrations of the migrating atom with its first (1st column) and second (2nd column) neighbor accounting for the full anharmonic interactions (1st row) and quasiharmonic interactions only (2nd row). The trajectory in **b** and the distribution functions in **c–f** correspond to MTP molecular dynamics at 3400 K and at the corresponding lattice constant of 3.24 Å.

with the sequence of bulk to vacancy to transition state. As a consequence, a more negative free energy contribution can be expected for both vacancy formation and migration Gibbs energies as indeed seen in Fig. 3.

**Collective motion of the migrating atom and its neighborhood**
For the vacancy formation process, the physical origin of anharmonicity is well understood within the local Grüneisen theory[23]. The here developed TSTI approach allows us to also access the full vibrational space of the transition state of vacancy migration. To understand the underlying physics of the strong anharmonicity, we collect long-time molecular-dynamics trajectories for $3 \times 10^5$ steps (1.5 nanoseconds) and examine the relative motion of the migrating atom with respect to its first and second neighbors. The geometrical relations are shown in Fig. 5a. In Fig. 5c–f, we plot the projection of the displacements of the 1st (2nd) neighbor atom onto $\mathbf{v_1}$ ($\mathbf{v_2}$) *versus* the projection of the

migrating atom onto $-\mathbf{v_1}$ ($\mathbf{v_3}$). The vectors $\mathbf{v_1}$ and $-\mathbf{v_1}$ lie opposite to each other along the line connecting the migrating atom and the 1st neighbor atom. The vectors $\mathbf{v_2}$ and $\mathbf{v_3}$ are directed towards the vacancy center ($\langle 111 \rangle$ direction).

As seen from Fig. 5, the full vibrations ((c) and (d)) and the harmonic approximation ((e) and (f)) describe different scenarios in terms of the distribution of the relative motion. For the interaction with the first neighbor, we observe the breakdown of harmonic symmetry due to Pauli repulsion as found previously for bulk anharmonicity[51]. The motion of the two atoms towards each other is energetically disfavored (suppressed distribution in quadrant I in (c)), whereas the motion away from each other is favored (increased distribution in quadrant III). This distinctive feature of the fully anharmonic potential cannot be described by any harmonic potential since the latter will always enforce the same energy profile whether two atoms move to or away from each other.

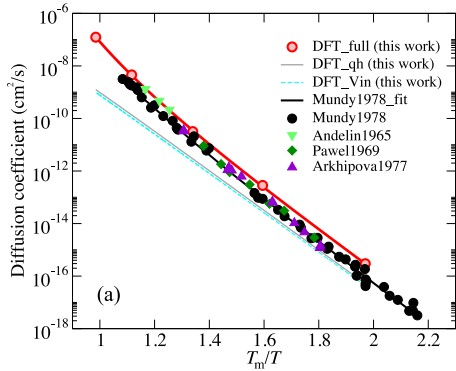
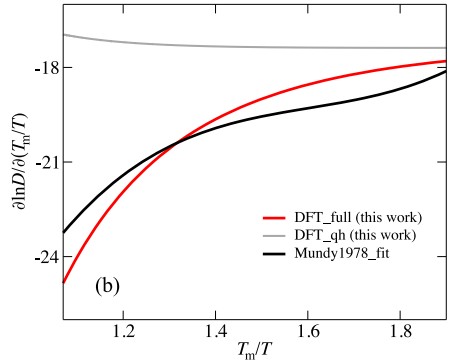

**Fig. 6 | Non-Arrenius self-diffusion of tungsten. a** Arrhenius plots of self-diffusivity in BCC tungsten calculated with the proposed ab initio machine-learning TSTI approach with all finite-temperature excitations taken into account (red circles and lines) on the homologous temperature scale in comparison to experimental data from Mundy et al.[12], Andelin et al.[55], Pawel et al.[56], and Arkhipova et al.[57]. **b** A comparison of the temperature-dependent slopes of the Arrhenius plots in **a**.

For the interaction of the second neighbor with the migrating atom (Fig. 5d), we observe likewise strong anharmonicity. The distribution function is very skewed along the positive $\mathbf{v_3}$ vector. This means that the motion of the second neighbor toward the vacancy is energetically preferred. Effectively, the second neighbor is drawn into the vacant site. Such preferential inward vibrations of atoms next to a vacancy were reported for vacancy *formation*[23]. With the presence of the migrating atom at the saddle point, interestingly and not reported so far, such an anharmonic inward motion of the 2nd neighbor atom is correlated with the motion of the migrating atom: it is particularly favored when the migrating atom moves away from the vacancy (toward the other vacant site), as is shown by the non-symmetric distributions in quadrant I and II of Fig. 5d and highlighted by the yellow arrows in the real-time MD trajectory depicted in Fig. 5b. These anharmonic characteristics are completely absent in the harmonic distributions. Note that such a concerted motion of a group of atoms might be favored for shallow potential landscapes, as it is anticipated for diffusion in melts[52], amorphous materials[53], or even for grain boundary diffusion[54].

### Non-Arrenius self-diffusion of tungsten

The Arrhenius plot of the calculated self-diffusion is shown in Fig. 6a as a function of the inverse homologous temperature, and compared with the most reliable experiments[12,55–57]. We use the DFT-PBE predicted melting point of 3349 K[58] for the present DFT results and the experimental melting point of 3687 K from the CRC Handbook[59] for the experimental data. The Arrhenius plot on the inverse homologous temperature scale compensates for the significant discrepancy (more than 300 K) between the DFT (GGA-PBE) predicted melting point and the experimental melting point for BCC W, as discussed in ref. 40. A corresponding plot on the absolute temperature scale is provided in the Supplementary Information.

The strong non-linear temperature dependence of the diffusion coefficient is revealed in Fig. 6b. The full DFT results agree very well with experiment. In contrast, the diffusivity derived from the (quasi)harmonic approximation (hTST; including the simplified Vineyard formula) yields an incorrect linear behavior. As the present DFT quasiharmonic results take the thermal expansion effect fully into account, the lattice expansion, previously proposed to explain the non-Arrhenius behavior[60], plays a negligible role in the high-temperature non-Arrhenius diffusion of W. We conclude instead that the non-Arrhenius self-diffusion of tungsten originates from strong anharmonicity in both the formation and migration of monovacancies. Since the calculations are based on the single monovacancy mechanism, the consistent curvature of the full DFT results with experiment indicates that the alternative di-vacancy mechanism plays a minor role in the non-Arrhenius diffusion of tungsten. A detailed analysis of the di-vacancy hypothesis, likewise leading to the exclusion of this mechanism, is provided in the Supplementary Information. We attribute the remaining discrepancy of the computed diffusion coefficient with experiment (Fig. 6a and Supplementary Fig. 5) to the employed exchange-correlation functional and experimental uncertainties.

### General applicability of TSTI

As the key ingredients of TSTI, i.e., the stabilization of the dynamical matrix and the MTP forces, are *system-independent* in their construction, the TSTI approach can be applied to diffusion in metals and alloys of any crystalline structure. Although we have focused in this work primarily on the self-diffusion of tungsten, applications of TSTI to more complex systems such as five-component high-entropy alloys (HEAs) are on their way. Previous studies[61,62] have demonstrated that MTPs are efficient and accurate in describing the vibrational phase space, even for such complex multicomponent systems. The achievable root-mean-square errors of only 2–3 meV/atom are the foundational premise for the success of direct upsampling.

To prove the general applicability of TSTI for complex HEAs with different crystalline structures other than BCC W, we show preliminary results in the Supplementary Information for a new category of hexagonal close-packed (HCP) HEAs, $Al_{15}Hf_{25}Sc_{10}Ti_{25}Zr_{25}$, which exhibit promising mechanical properties[63] and intriguing defect and diffusion properties in experiment and theory[64–67]. An accurate MTP with an RMSE of 2.5 meV/atom has been developed. The resulting well-behaved $\langle E_s^{full} - E^{qh} \rangle_\lambda$ curve (Supplementary Fig. 9a) and the stable MD trajectory of the migrating atom (Zr) at the transition state (Supplementary Fig. 9b) substantiate the broad applicability and the robust performance of the TSTI approach. A complete simulation study of the diffusion in HEAs with DFT accuracy requires an increased computational effort due to the additional chemical complexity. Statistical sampling in both configurational and vibrational degrees of freedom must be performed and converged to obtain reasonable diffusivities. Such an investigation goes beyond the scope of the present study and is the focus of a separate project.

## Discussion

Utilizing a bespoke machine-learning potential, we have developed an efficient ab initio framework, the *transition state thermodynamic integration* (TSTI) approach, to accurately compute the Gibbs energy of the transition state. The inherent dynamical instability—so far limiting the use of thermodynamic integration—has been successfully overcome.

We have applied the proposed framework to predict the temperature-dependent vacancy migration and formation Gibbs energies of BCC W, revealing strong anharmonic atomic vibrations. The resulting migration and formation Gibbs energies exhibit a significant non-linear temperature dependence. Through the analysis of long-time molecular-dynamics trajectories, we have related the strong anharmonicity to a non-symmetric relative motion of the first and the second neighbor atoms with respect to the migrating atom at the transition state. The predicted temperature-dependent self-diffusion coefficient shows clearly a non-Arrhenius curvature and agrees remarkably well with experimental data on the inverse homologous temperature scale. In contrast, the generally applied quasiharmonic approximation cannot explain the observed non-Arrhenius behavior.

The developed framework opens new avenues in probing the diffusion behavior with ab initio accuracy, including transition states in the whole temperature range. It allows us to understand observed diffusion anomalies and resolve long-standing and highly debated problems. Diffusion properties of metastable and unstable phases become computable with density-functional-theory accuracy, which is of paramount importance for consistent and reliable development of (CALPHAD-type) mobility databases with end-members for which no experimental data exist (like BCC Al or Ni for the assessment of mobilities in the B2 NiAl phase relevant to Ni-based superalloys). In particular, for dynamically unstable phases where the relaxation of vacancy-containing structures and also the nudged-elastic-band method become infeasible due to the instability, the present computational framework, which combines TSTI with direct upsampling, provides a unique advantage to treat such phases if they can be dynamically stabilized in molecular dynamics at elevated temperatures. Once a proper reference potential with known free energy is available, TSTI from the reference potential to the full vibrational potential (moment tensor potential (MTP) in this work) can be applied to obtain the anharmonic free energy contribution, followed by direct upsampling to achieve the full density-functional-theory accuracy. Possible references are, for example, the effective harmonic potential or the Einstein solid for the stable end-points of diffusion[68] and a migration path interpolated from stable reference elements for the transition state[60].

With the increasing importance of diffusion data in developing advanced materials, accurate theoretical prediction of diffusivities is highly desirable. As the introduced methodology is general, it can be extended beyond vacancy-mediated diffusion to transition-state Gibbs energies of other diffusion mechanisms. We have shown that it can be applied to complex alloy systems such as high-entropy alloys where the Gibbs energy contribution to diffusion from the configurational degree of freedom also shows a non-Arrhenius behavior[67]. Stabilized molecular-dynamics trajectories for the migrating atom at the transition state in a five-component HCP high-entropy-alloy system support the applicability of TSTI. The introduced and verified TSTI approach is straightforward to implement in a standard thermodynamic-integration code, and it can be thus suggested for community usage and versatile testing in other applications, particularly in studies of diffusion-controlled processes. As an ultimate goal, a comprehensive physics-informed accurate ab initio diffusion database can be expected.

## Methods
### General ab initio free energy approach for defects
The temperature-dependent Gibbs energies of vacancy formation and migration for pure metals are expressed as

$$G_{\text{form}}(T) = G_{\text{vac}}(T) - \frac{N-1}{N} G_{\text{bulk}}(T) \tag{12}$$

and

$$G_{\text{mig}}(T) = G_{\text{TS}}(T) - G_{\text{vac}}(T), \tag{13}$$

where $G_{\text{bulk}}$, $G_{\text{vac}}$, and $G_{\text{TS}}$ are the Gibbs energies of the perfect bulk, vacancy and transition-state supercell, and $N$ is the number of atoms in the perfect bulk supercell. At the equilibrium volume of a given temperature (zero pressure), the Gibbs energy and the Helmholtz energy coincide. To account for the various relevant temperature-dependent thermal excitations, the Helmholtz energy is adiabatically decomposed as[26]

$$F(V_{\text{eq}}(T), T) = E_{0K}(V_{\text{eq}}(T = 0\,\text{K})) + F^{\text{qh}}(V_{\text{eq}}(T), T) \\ + F^{\text{ah}}(V_{\text{eq}}(T), T) + F^{\text{el}}(V_{\text{eq}}(T), T), \tag{14}$$

where $E_{0K}$ denotes the 0 K total energy, $F^{\text{qh}}$ and $F^{\text{ah}}$ the contribution from non-interacting phonons (quasiharmonicity) and from *explicit* phonon-phonon interactions (anharmonicity), respectively. Further, $F^{\text{el}}$ refers to the electronic free energy with the impact of thermal vibrations taken into account. All free energy contributions correspond to the equilibrium volume $V_{\text{eq}}(T)$ at temperature $T$ according to the thermal expansion.

### Quasiharmonic free energy $F^{\text{qh}}$
The quasiharmonic free energy is calculated by applying the Bose-Einstein statistics to populate the phonon energy levels, i.e.,

$$F^{\text{qh}} = \sum_i^{3N} \left\{ \frac{\hbar\omega_i}{2} + k_B T \ln\left[ 1 - \exp\left( -\frac{\hbar\omega_i}{k_B T} \right) \right] \right\}, \tag{15}$$

with the reduced Planck constant $\hbar$ and phonon frequencies $\omega_i$. The finite-displacement supercell approach[69] is used to extract phonon frequencies, as implemented in phonopy[70].

For the transition state, the integration of phonon modes in Eq. (15) excludes the single imaginary mode along the reaction path, as described in the conventional harmonic transition state theory (hTST)[27].

### Explicitly anharmonic free energy $F^{\text{ah}}$
**Thermodynamic integration.** The explicitly anharmonic free energy $F^{\text{ah}}$, i.e., the difference between the full vibrational free energy and the quasiharmonic free energy, can be extracted by applying a thermodynamic integration (for bulk and vacancy configuration) or the proposed *transition state thermodynamic integration* (TSTI; for the transition state) along a predefined thermodynamic pathway between the quasiharmonic potential, $E^{\text{qh}}$, and the full vibrational potential, e.g, the moment tensor potential (MTP) $E^{\text{MTP}}$ in this work. In practice, the thermodynamic path is usually established by a linear interpolation, i.e.,

$$E_\lambda = (1 - \lambda) E^{\text{qh}} + \lambda E^{\text{MTP}}, \tag{16}$$

where $\lambda$ is a coupling parameter with a value between 0 and 1. For different $\lambda$, the atomic movement in the MD simulations is driven by forces described by either $E^{\text{qh}}(\lambda = 0)$ or $E^{\text{MTP}}(\lambda = 1)$ or a linear mixing of both $E_\lambda (0 < \lambda < 1)$. The anharmonic free energy then reads

$$F^{\text{ah}} = \int_0^1 d\lambda \left\langle \frac{\partial E_\lambda}{\partial \lambda} \right\rangle_\lambda = \int_0^1 d\lambda \left\langle E^{\text{MTP}} - E^{\text{qh}} \right\rangle_\lambda + \Delta F^{\text{up}}, \tag{17}$$

where $\langle \dots \rangle$ refers to the thermal average and the remaining minor free energy adjustment $\Delta F^{\text{up}}$ required for full DFT accuracy can be obtained from direct upsampling[62].

**Direct upsampling.** The free energy difference between MTP and DFT can be calculated in a perturbative manner using the free-energy perturbation expression, as given by,

$$\Delta F^{\text{up}} = -k_B T \ln \left\langle \exp\left(-\frac{E^{\text{DFT}} - E^{\text{MTP}}}{k_B T}\right)\right\rangle_{\text{MTP}}, \qquad (18)$$

where $k_B$ is the Boltzmann constant and $E^{\text{DFT}}$ and $E^{\text{MTP}}$ are DFT and MTP energies of molecular dynamics snapshots obtained from MTP trajectories.

## Electronic free energy including thermal vibration coupling effects

Due to the accurate MTP forces as compared to DFT (Fig. 2b), the free energy perturbation theory can also be applied to capture the free energy difference between DFT with and without electronic excitations utilizing the MD snapshots generated by the MTP. The electronic free energy, including the coupling effects, can be taken into account by setting the electronic temperature in the DFT calculations to the respective molecular dynamics temperature[36], i.e.,

$$F^{\text{el}} = -k_B T \ln \left\langle \exp\left(-\frac{E_{\text{el}}^{\text{DFT}} - E^{\text{DFT}}}{k_B T}\right)\right\rangle_{\text{MTP}}. \qquad (19)$$

## Moment tensor potentials

**Potential form.** To boost up the TSTI calculations, we employ the moment tensor potential (MTP)[35]. MTP utilizes moment tensor descriptors to characterize the local atomic environment $n_i$ (see Supplementary Fig. 2):

$$M_{\mu,\nu}(n_i) = \sum_j f_\mu(|r_{ij}|, z_i, z_j)\underbrace{\boldsymbol{r}_{ij} \otimes \dots \otimes \boldsymbol{r}_{ij}}_{\nu \text{ times}}, \qquad (20)$$

where the radial function $f_\mu$ encapsulates information about relative atomic distances ($|\boldsymbol{r}_{ij}|$) and chemical types ($z_i, z_j$), while the angular part $\boldsymbol{r}_{ij} \otimes \dots \otimes \boldsymbol{r}_{ij}$ represents the moments of inertia. The radial function is expressed as

$$f_\mu(|\boldsymbol{r}_{ij}|, z_i, z_j) = \sum_k c_{\mu,z_i,z_j}^k T_k(|\boldsymbol{r}_{ij}|)(R_{\text{cut}} - |\boldsymbol{r}_{ij}|)^2, \qquad (21)$$

where $c_{\mu,z_i,z_j}^k$ are the radial coefficients to be fit, $T_k(|\boldsymbol{r}_{ij}|)$ are the Chebyshev polynomials of order $k$, and $R_{\text{cut}}$ is the cutoff radius (red arrow in Supplementary Fig. 2). Here, we used a cutoff radius of 5.2 Å covering the 1NN, 2NN, and 3NN neighbor shells of BCC W for all equilibrium lattice constants from 0 K to 3400 K, which is sufficient for describing the local environment. For the angular part, in order to determine the basis set of the potential, one has to set up the so-called "maximum level of moments", $\text{lev}_{\text{max}}$, defined in ref. 71 using a notation of "$N$g" with $N$ being the value of $\text{lev}_{\text{max}}$. Generally, the higher the maximum level of moments, the larger the number of basis functions and fitting parameters the MTP model possesses. In this work, a high-level "24g" MTP model ($\text{lev}_{\text{max}} = 24$) containing 912 fitting parameters for the unary tungsten system was selected to ensure accuracy in describing the vibrational space. The potential energy is expressed as the sum of contributions from all local environments, i.e., $E^{\text{MTP}} = \sum_i V(n_i) = \sum_i \sum_\alpha \xi_\alpha B_\alpha(n_i)$, where $B_\alpha$ denotes basis functions derived from contractions of the moment tensors, and $\xi_\alpha$ represents linear fitting parameters.

**Training dataset and active learning.** The DFT dataset for training the MTP was obtained from *NVT* MD snapshots at 3600 K and at four different lattice constants (3.206 Å, 3.227 Å, 3.248 Å, and 3.268 Å) to ensure that a sufficiently large vibrational space can be explored and learned by the MTP.

An iterative approach according to the "active learning" scheme detailed in ref. 71 was used to automatically select new configurations to be added in the training set. Based on the D-optimality criterion, an extrapolation grade $\gamma$ can be calculated for each newly-generated MD snapshot using the so-called maxvol algorithm[71]. The extrapolation grade is a measure of the "novelty" of the new structures compared to the ones already in the training set. From a mathematical point of view, extrapolation by the MTP occurs if $\gamma$ is greater than unity[71]. The higher the extrapolation grade the severer is the extrapolation by the MTP. During the training process, we used the selection threshold $\gamma_{\text{select}} = 1.1$, i.e., the energies and forces of new configurations with an extrapolation grade larger than 1.1 were calculated by DFT and added to the training set.

The active learning scheme started with an initial MTP trained with 200 bulk MD structures. New MD runs with the MTP obtained in the previous step were performed from which new bulk structures, vacancy structures, and transition state structures were selected and added to the training set. The MTP was then re-trained with the updated training set. The "active learning" procedure stopped when addition of new structures during the MD runs became statistically insignificant. The resulting final training database contained 806 bulk structures, 872 vacancy structures, and 913 transition state structures (2591 in total).

## Computational details

**DFT calculations.** The total energies and forces of configurations selected for training the machine-learning potential as well as the energies of the upsampling snapshots were obtained from DFT calculations. The GGA-PBE exchange-correlation functional was used for all the DFT calculations. A previous ab initio study[23] on temperature-dependent vacancy formation Gibbs energies of Al and Cu showed an outstanding performance of PBE compared to other functionals. The PBE functional was also applied to refractory BCC elements including W in ref. 40, and a very good prediction of thermodynamic properties up to the melting point was demonstrated, in particular on the homologous temperature scale. In ref. 40, the Gibbs energy of vacancy formation was also calculated, once at 0 K and additionally at one high-temperature point indicating a strong temperature dependence. In another recent study[58], the melting point of BCC W was calculated with the PBE functional, which enables a comparison with experimental data on the homologous temperature scale. The PBE functional is thus well suited for our present study, and our simulations nicely complement the available literature with the full temperature dependence of formation and migration Gibbs energies for BCC W.

The PAW potential with $5d^4 6s^2$ valence orbitals from the VASP potential database was used. The plane wave cut-off was set to 250 eV. Supercells based on a $4 \times 4 \times 4$ expansion of the conventional BCC unit cell were used (128 atoms for bulk structures and 127 atoms for the vacancy and transition state). A $\Gamma$-centered Monkhorst-Pack[72] $k$-point mesh of $3 \times 3 \times 3$ was used for all except the separate quasiharmonic calculations where a mesh of $9 \times 9 \times 9$ was applied for convergence. The convergence criterion for the self-consistent electronic loop was specified to be $10^{-5}$ eV ($10^{-6}$ eV for the quasiharmonic calculations).

**Nudged elastic band calculations.** The transition-state structure $\{\boldsymbol{R}_I^{\text{sp}}\}$ at a given volume was obtained from DFT-based climbing image nudged elastic band (NEB)[38] calculations implemented in VASP with 11 intermediate images. The spring constant between the images was set to $-5$ eV/Å$^2$. The NEB ionic relaxation with the quasi-Newton algorithm was stopped when all the forces were smaller than 0.05 eV/Å. The total energies of all intermediate images, together with the two end points, were fitted to a force-based cubic spline interpolation using the

tangential forces as input to accurately locate the saddle point, as implemented in the VTST-scripts[73].

**Thermodynamic integration.** Utilizing the highly-optimized machine learning potential, at each temperature and volume ($T,V$) point, the thermodynamic integration (TI) from the quasiharmonic potential to the full vibrational moment tensor potential could be efficiently performed using eight $\lambda$ values, i.e., 0, 0.2, 0.4, 0.6, 0.8, 0.9, 0.95, and 1, followed by a tangent fit. All TI MD simulations were run until a standard error of below 0.05 eV/cell for the resulting free energy was reached. The Langevin thermostat was used with a friction parameter of 0.01 fs$^{-1}$ and a time step of 5 fs.

**Direct upsampling.** For the direct upsampling procedure, the number of MTP MD snapshots calculated by DFT for statistical averaging ranged from 360 to 720, depending on the volume and temperature. These numbers are more than one order of magnitude larger than the general upsampling procedure for the bulk (20 ~ 30 snapshots for an accuracy of 1 meV/atom). The resulting convergence allows us to reach an accuracy of below 0.2 meV/atom. This very high accuracy per atom translates into an accuracy of about 20 meV/defect.

## Data availability
All data supporting the findings of this study are available in the article and its Supplementary Information file. All key data files required to reproduce the results and the source data presented in graphs within the Figures and the Supplementary Figs. are available on DaRUS (https://doi.org/10.18419/darus-4564).

## Code availability
The authors declare that the custom codes/software for the transition state thermodynamic integration are available from the corresponding author upon request.

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

## Acknowledgements

X.Z. thanks Wenchuan Liu for the quasiharmonic calculations and Dr. Axel Forslund for the fruitful discussions. Financial support from the European Research Council (ERC) under the European Unions Horizon 2020 research and innovation program (Grant Agreement No. 865855)

is gratefully acknowledged. S.D. and X.Z. acknowledge funding from the Deutsche Forschungsgemeinschaft (DFG) via the research projects DI 1419/24-1 and ZH 1218/1-1. X.Z. and B.G. acknowledge support by the state of Baden-Württemberg through bwHPC and the DFG through grant No. INST 40/575-1 FUGG (JUSTUS 2 cluster) and by the Stuttgart Center for Simulation Science (SimTech).

## Author contributions

X.Z. designed and conducted the simulations, S.D. checked the comparison with the experiment, X.Z. and B.G. analyzed the results and wrote the manuscript. All authors reviewed the manuscript.

## Funding

## Competing interests

The authors declare no competing interests.
