## [Transparent Peer Review file · Nature Communications]

Ab initio machine-learning unveils strong anharmonicity in non-Arrhenius self-diffusion of tungsten

Corresponding Author: Dr Xi Zhang

Version 1:

Reviewer comments:

Reviewer #1

(Remarks to the Author)

The authors present a study utilizing machine-learning assisted ab initio simulations to calculate the mono vacancy diffusion in bulk tungsten, which reveals a non-Arrhenius temperature dependence. From explicitly projecting the atomic trajectories on to different migration modes, they attributed such non-Arrhenius behavior to the collective motion with the neighboring atom. They also specified the contribution from the electronic excitations and made a comparison with the experimental data. Finally, the authors discussed the possibility of extending such a framework for multi-component high entropy alloys. Still, there are a number of questions and issues however that I believe the authors should address before the manuscript proceeds towards publication, which I will elaborate on details below. In my opinion, the manuscript has great potential but without these discussions, the impact of the work could be limited.

1) General conclusion

1.1) The authors claim that the method developed in this work captures the non-Arrhenius feature and agrees remarkably well with experimental data. However, looking at the data points on the same temperature scale (Fig. S2), there is a difference of two orders of deviating from the measured diffusion coefficients, even larger than the DFT_qh. It would be necessary to either compare with other theoretical results or to quantify and identify the source of the deviation before such a conclusion can be drawn.

1.2) A related question to 1.1, thermal expansion can serve as another manifestation or source of anharmonicity, as suggested in e.g. Phys. Rev. Materials 6, 063402 (2022) where by accounting for the thermal expansion alone a great part of the non-Arrhenius diffusion is well captured. The authors only referred to Ref.[39] which is not the same system. Can the authors evaluate the thermal expansion coefficients obtained from their method and compare them with existing data?

1.3) A significant aspect of the outlook lies in extending this method to metastable and unstable phases with DFT accuracy. The DFT-based calculations usually suffer from not only the imaginary phonon frequencies, but also the intractability of relaxing the vacancy structure in e.g. high-temperature bcc phases of Ti and Zr, etc. Can the authors elaborate on how their method will do in such systems?

2) On the electronic excitations

The authors compared the eDOS at 3000K for different structures, and revealed its strong impact on the vacancy formation and migration energies. Albeit the data plotted, the physical connection appears weak.

2.1) What is the lattice temperature during the MD simulation? How large is the mean deviation of the eDOS of the 120 MD snapshots, specifically in comparison with the difference between bulk/vacancy/transition structures? Does the DOS of vacancy and bulk structures have distinguishable differences (e.g. in [e.g. Phys. Rev. B 57, 11184 (1998)]) at 0 K and 3000 K?

2.2) In Fig. 2b, the migration energy for the "qh+ah+el" data point seems to be larger than that of the "qh+ah". Can the authors comment on this? The underlying physical connection between the electronic temperature (if it is simply the smearing factor

in the MD calculations) and the vacancy formation/migration energies is not very clear.

3) On the collective migration

The projection of the atomic trajectories onto different migration modes provides an interesting perspective regarding a general understanding of anomalous self-diffusion behaviors in many systems.

3.1) In the long-time molecular dynamics simulation specifically, how is the lattice degree of freedom controlled? What is the major difference in performing calculations with "full" vibrations vs. "harmonic" vibrations?

3.2) What is the color coding in Fig.4(b-e)? Is it corresponding to the time evolution, or is it some kind of histogram? Other than the projected intensity on different migration vectors, can the authors show some real-time snapshots like for example in Refs. [Nat. Geosci. 10, 312 (2017), Phys. Rev. Lett. 123, 105501 (2019)] to illustrate the collective diffusion?

4) Miscellaneous

I found inconsistency in e.g. figure references between the SI and the main text and suggest the authors to give a careful check.

Reviewer #2

(Remarks to the Author)

In many pure metals, it has been observed that the measured diffusivity due to vacancy-atom exchanges deviates from Arrhenius behavior. More specifically, experiments are rationalized by assuming that the activation energy, Q , and the diffusivity prefactor D_0 are constants, independent of temperature.

Deviations from Arrhenius behavior have been typically ascribed to the presence of defects other than mono-vacancies as the mediators for atomic substitutional exchange. Di-vacancies, for example, have been invoked to explain deviations from Arrhenius behavior. The authors point out that other explanations have been invoked, including contributions due to anharmonicity. In this manuscript the authors address the problem using a highly novel and efficient computational method to predict the different factors that, together, control the vacancy-mediated diffusion in metals. They proceed to apply the framework to investigate the supposed diffusion anomaly in Tungsten and further demonstrate the utility of the framework to predict the diffusivity in hcp-based high entropy alloys.

On the methodological front, this paper is impeccable. The authors clearly explain the limitations of current DFT-based approaches to predict the thermodynamics of non-equilibrium states, such as those corresponding to a diffusing atom 'sitting' a top of an energy barrier as it transitions from one substitutional site to another through exchanging places with a vacancy. To address this issue the authors employ a state-of-the-art machine learning potential to attain energetic predictions at the DFT-level of precision. They then use highly advanced thermodynamic integration tools to estimate the temperature dependence of the vacancy formation and migration Gibbs free energies, accounting for quasi-harmonic, electronic and anharmonic effects. The anharmonicity in this case is assumed to arise (at least in the case of the migration Gibbs energy) from the non-harmonic shape of the crystal field along the direction of atomic migration.

While the method used represents the state of the art and the culmination of the trajectory towards increased fidelity in the estimation of thermodynamic properties by this group (among the leaders in the field), the results and interpretation fall short. Below some of the most salient issues are presented:

Perhaps the most evident problem is the fact that the authors are using an advanced theoretical approach to test the hypothesis that non-Arrhenius behavior in the diffusion of W is due to anharmonic contributions to the vacancy's migration and formation free energies. However, this test does not consider the commonly invoked alternative hypothesis, namely, that di-vacancies may be responsible for (at least some of) this deviation from Arrhenius behavior. Trying to validate a theory at the DFT level with experiments is very challenging (more on this later), so a first step would be to carry out an analysis assuming that di-vacancies are present and then try to come up with arguments based on theoretical grounds and some type of comparison with experiments in order to establish the validity of either of the two competing hypotheses (there could be more explanations).

Another problem has to do with the way the authors try to establish the correctness of their predictions by comparing against a sparse set of experimental results. The authors seem to extrapolate (Fig 2) the values of the high-temperature migration and formation energies to low temperature, claiming that this is analogous to approaches whereby experimental temperature series have been extrapolated to low temperatures. A first question is then why not comparing the measurements done at high temperatures with the predictions carried out at similar temperatures? It is not clear why the authors chose to only compare extrapolations to low temperatures when they produce high-temperature predictions and experiments (as claimed by these authors) have been carried out at similarly high temperatures.

Another question that arises when examining their analysis in Fig. 2. Specifically, they choose to extrapolate to low temperatures their calculations considering anharmonic effects. Yet, at least in the case of the vacancy migration energy, it seems that extrapolating from the 'qh+el' approximation may also yield results that would be comparable to experimentally

measured values, within the uncertainty of the experiments.

This last point brings another question: the authors are presenting their computational predictions in a deterministic manner and no uncertainty bounds are presented in their calculations. This is definitely not correct as surely there must be a number of factors that would contribute to uncertainty in the predictions, including calculations setups, pseudopotential choice, uncertainties arising from numerical uncertainties in their thermodynamic integration, uncertainties due to imperfect training of the ML potential, etc. Over the past decade there have been a number of works that have pointed out the importance that accounting for uncertainty in computational materials science predictions if one is to compare with experiments.

One last problematic aspect of the paper has to do with Fig. 5. In Fig. 5a, the authors present a number of temperature series corresponding to their own calculations under different levels of approximation as well as some experimental data. From their results, it is evident that their predictions where they account for all the contributions suggest a non-Arrhenius behavior of the diffusivity of tungsten. The experiments presented tend to be consistent with each other with some scattering.

While there may seem that there is some deviation from linearity in $\log - 1/T$ coordinates, I believe that the analysis presented in Fig. 5b to reinforce the authors' argument for the importance of anharmonicity in the diffusion of tungsten is not really as strong as they suggest. The biggest issue is that they seek to make their case by taking a derivative of their predictions as a function of temperature and do the same on a fit to an experimental dataset. While I accept that their own data exhibits a deviation from "0" that suggests that the prefactor and migration energies may be temperature-dependent, the same cannot be said, unquestionably, in the case of the experimental dataset. After all, the temperature dependence of these quantities is determined by the form of the function used to fit the data. If, for example, a linear fit would have been used, then there would not be any possibility for inferring a non-Arrhenius behavior in the experimental data. In fact, while all the data is in agreement, there is sufficient scatter in the experimental data to preclude any kind of certainty in the use a specific function or other when fitting the dataset.

At the introduction of their manuscript, the authors correctly state that (paraphrasing), extraordinary claims require extraordinary evidence. Unfortunately, in this case, while the authors do present some plausible evidence for non-Arrhenius diffusivity in tungsten, the case is by no means strong. In fact, the lack of rigorous uncertainty/statistical analysis makes it impossible to assert any level of confidence in their conclusions. It is not clear, to me at least, that the authors have made their case with 100% certainty. It is probably not the case that this certainty is 0%. It must therefore be somewhere in between but without proper analysis it is not possible to ascertain the correctness of their predictions. These deficiencies in no way detract from the excellent and much-needed methodological improvements in first principles predictions.

Reviewer #3

(Remarks to the Author)

The manuscript titled "Ab initio machine-learning unveils strong anharmonicity in non-Arrhenius self-diffusion of tungsten" introduces a numerical framework integrating Machine Learning, Density Functional Theory (DFT) calculations, and canonical sampling methods. This framework adeptly calculates the free energy associated with the formation and migration of a mono-vacancy in tungsten. By directly estimating these quantities, the authors determine the temperature-dependent self-diffusion coefficient of tungsten. They conclude that the anomalous high-temperature behavior of this coefficient, deviating from the Arrhenius model, is due to anharmonic and electronic contributions in the free energy of this basic defect.

The theoretical and numerical literature on this subject has been contentious, particularly regarding the cause of the experimentally observed non-standard Arrhenius behavior of the self-diffusion coefficient at high temperatures. Many studies in the late '70s and early '80s investigated the origin of this phenomenon. Anharmonicity was proposed but integrated into the general thermodynamic model without support from atomistic calculations. In the '90s and late 2000s, the prevailing view was that the curvature in the self-diffusion coefficient stemmed from di-vacancies rather than mono-vacancies. This is where this study excels; the authors attempt to clarify this issue. The results align with experimental data trends, although they consistently overestimate the self-diffusion coefficient.

However, despite the positive aspects, I have significant reservations about this study. Although the concept of a completely coupled framework is intriguing, the central method for sampling the saddle point with ab initio accuracy seems too far-fetched and mathematically imprecise. The authors possess extensive experience in sampling minima, but it appears that saddle points (a parabolic energy surface with one unstable mode of the Hessian) fall outside their expertise. The free energy in the current framework is inevitably biased. It may veer towards either the lower or upper limit of the true value, but there is a systematic error in the evaluation of sampling averages. This critical issue is not addressed in the paper, and I consider it a central flaw that could affect the final results. Unless this issue is resolved, I cannot envision a path to publication for this paper. To put it succinctly, in its current form, I do not recommend the publication of the paper. From a mathematical standpoint, I find the results to be irrelevant.

In this method the quasiharmonic decomposition of the Hessian is used in order to compensate 0 K Hessian unstable mode. Then by fixing the the interatomic force as a mix between the DFT quasiharmonic and MTP force fields by linear interpolation with some tricky projection in the unstable direction, the TI integration of performed in this completely parabolic as-for-a-minimum energy landscape. Here are two major sources of bias:

The first source of bias is the mixing of forces. Based on the equations presented in the section "Transition state thermodynamics integration," it is unclear how the authors carry out the sampling method. The text and equations create

confusion between classical TI (equation 3) and equation 7, where forces are mixed or averaged through a mixing parameter. The authors do not clarify the sampling dynamics of the external parameter λ (in equation 7), which unfortunately has the same notation but a completely different meaning compared to equation 3.

The second source of bias arises from the authors' use of a biasing quasi-harmonic method to circumvent the impossibility of direct sampling at the saddle point. This approach is based on the assumption that the flow direction at the saddle point (which is identified as the Hessian eigenvector at 0K) exhibits low anharmonicity and its direction remains unchanged vectorially at any temperature of the sampling. This hypothesis is unverified, and other studies, even in low-dimensional and simple energy landscapes, easily demonstrate the contrary. The pathways at 0K are not the same as those at finite temperatures, and the authors do not account for this discrepancy in their sampling approach, unlike the treatment proposed by D. Branduardi, F. L. Gervasio, and M. Parrinello in "From A to B in Free Energy Space," J. Chem. Phys. 126, 054103 (2007).

The bias in the free energy calculation is not quantified, which is a significant concern. The authors' statement that "the unstable mode, which drives the saddle-point configuration to equilibrium, does not contribute to the vibrational free energy of the transition state and can be modified to a stable one without altering the free energy" demonstrates a fundamental misunderstanding of sampling methods. Although the unstable mode does not contribute to the free energy after thermodynamic integration, the sampling is skewed by the quasi-harmonic potential. This major methodological flaw raises serious doubts about the analysis of the results.

The authors present their methodology as a significant advancement in the field. However, the manuscript's description of this methodology leaves substantial uncertainties about its validity. There is a notable lack of comparison with established methods from the literature, such as the string method referenced in paper 32, especially for simpler cases. Given these critical issues, I am unable to endorse the publication of this paper. The unaddressed biases and methodological oversights significantly undermine the credibility and relevance of the findings. Therefore, I must recommend the rejection of this manuscript.

Minor points:

- For all equations involving F^{ah} , the quasi-harmonic free energy is missing (e.g., equation 2 and equation 16).
- There is a lack of data (such as potential design, fitting database, error analysis, etc.) provided for the used MTP machine learning potential, with the exception of RMSE for forces. For a review as Nature Com, where there is enough space for clarifying methods, supp. materials etc is quite surprising.
- Equation (18) presents thermodynamic perturbations between DFT and temperature DFT data on the MTP canonical ensemble. However, there is no data provided about distribution overlaps, even though MTP potentials are typically used in an active learning framework due to their energy landscape instability.
- the authors claim that their framework can be, or is being, used for High-Entropy Alloys (HEA). However, HEAs are only mentioned briefly in the main manuscript, and only a single result is provided in the supplementary material.

Reviewer #4

(Remarks to the Author)

I co-reviewed this manuscript with one of the listed reviewers

Version 2:

Reviewer comments:

Reviewer #1

(Remarks to the Author)

The authors have successfully addressed my concerns in the response letter and revised manuscript. The more detailed discussions on the contribution of electronic excitation is more clear and the outlook on how to generalize this method in unstable structure is insightful. The atomic trajectory is helpful for visualization as well. I would recommend this manuscript for publication.

(Remarks on code availability)

If possible, the authors are suggested to provide input file samples .

Reviewer #2

(Remarks to the Author)

The authors have address my concerns to my satisfaction.

(Remarks on code availability)

Reviewer #3

(Remarks to the Author)

Please find the report in the attached pdf file.

(Remarks on code availability)

Reviewer #4

(Remarks to the Author)

I co-reviewed this manuscript with one of the listed reviewers.

(Remarks on code availability)

Version 3:

Reviewer comments:

Reviewer #2

(Remarks to the Author)

I have looked carefully at the reviewers' objections to the present manuscript as well as the authors' response. I believe that the underlying reasons for the objections result from a misunderstanding of the thermodynamic integration procedure followed by the authors. The authors are using rigorous transition state theory to integrate out the unstable mode according to practice established over a long time, modified by their own state-of-the-art methods. It is thus my opinion that the authors are correct and therefore I do not think there are further issues with the present manuscript.

(Remarks on code availability)

Reviewer #3

(Remarks to the Author)

It seems that the authors remain steadfast in their positions. Please find the report in the attached pdf file. There, you will also find clear proof of the incorrectness of their method. Consequently, we have reached the same conclusion: while the present paper seems interesting, the method presented, in its current form, is simply incorrect. Although some aspects could potentially be improved or corrected, none of these improvements were evident in the authors' reply. Therefore, with regret, I must say that in its present form, I can only recommend rejection.

(Remarks on code availability)

Reviewer #4

(Remarks to the Author)

I co-reviewed this manuscript with one of the listed reviewers.

(Remarks on code availability)

Dear Reviewers,

We thank all four of you for the reports and the constructive comments. Following your comments and suggestions, we have thoroughly revised the manuscript and the Supplementary Material. In particular, besides the careful revision of the manuscript, we have significantly extended the Supplementary Material with extensive computational details and discussions including detailed explanations to the key concerns that you have raised. We fully agree that this information and the discussions are of great benefit to the general readers of Nature Communications.

A detailed response to each of the comments and the resulting revisions are given below. The main changes are highlighted in magenta in the revised manuscript, the Supplementary Material and this letter. We hope that the revised manuscript will be suitable for publication in Nature Communications.

With best regards,
Xi Zhang, on behalf of all co-authors

Contents

Responses to Reviewer #1	2
General conclusion	2
Thermal expansion	3
Vacancies in dynamically unstable phases	5
On the electronic excitations	7
On the collective migration	11
Miscellaneous	12
Responses to Reviewer #2	13
On the di-vacancy hypothesis	13
Arrhenius extrapolation	15
Uncertainty analysis	17
Non-Arrhenius behavior	17
Responses to Reviewer #3	20
Mixing of the forces	20
Migration pathway	21
Minor points	23
Response to Reviewer #4	26

Responses to Reviewer #1

General comment: The authors present a study utilizing machine-learning assisted ab initio simulations to calculate the mono vacancy diffusion in bulk tungsten, which reveals a non-Arrhenius temperature dependence. From explicitly projecting the atomic trajectories on to different migration modes, they attributed such non-Arrhenius behavior to the collective motion with the neighboring atom. They also specified the contribution from the electronic excitations and made a comparison with the experimental data. Finally, the authors discussed the possibility of extending such a framework for multi-component high entropy alloys. Still, there are a number of questions and issues however that I believe the authors should address before the manuscript proceeds towards publication, which I will elaborate on details below. In my opinion, the manuscript has great potential but without these discussions, the impact of the work could be limited.

Reply: We thank you for the appreciation of our work. Detailed point-to-point responses and revisions are given below.

General conclusion

Comment 1.1: The authors claim that the method developed in this work captures the non-Arrhenius feature and agrees remarkably well with experimental data. However, looking at the data points on the same temperature scale (Fig. S2), there is a difference of two orders of deviating from the measured diffusion coefficients, even larger than the DFT_qh. It would be necessary to either compare with other theoretical results or to quantify and identify the source of the deviation before such a conclusion can be drawn.

Reply: We need to stress that for a fair comparison of the calculated diffusivities with experiment, the large discrepancy (more than 300 K) between the DFT (GGA-PBE) predicted melting point (3349 K, reported in Ref. [59] of the manuscript [Phys. Rev. B 109, 094110]) and the experimental melting point (3687 K) should be taken into account. Correspondingly, for two systems with different melting temperatures, an adequate comparison of the diffusion rates in the Arrhenius plot should be done on the inverse homologous temperature scale, i.e., T_m/T , instead of the inverse absolute temperature scale $1/T$, in order to compensate for the discrepancy in the melting points. The error in the DFT-predicted melting point of tungsten stems from the selected GGA-PBE exchange-correlation (XC) functional which is, in general for refractory BCC elements, softer than the true potential. This leads to a systematic overestimation of thermodynamic properties such as the heat capacity and the thermal expansion and a systematic underestimation of the bulk modulus and the melting temperature when compared on the absolute temperature scale, as reported in [Phys. Rev. B 107, 174309 (2023)] for BCC V, Ta, Mo, and W. Consequently, an overestimation of the diffusivities on the absolute temperature scale can be expected. In contrast, a comparison on the homologous temperature scale shows a good agreement for various thermodynamic properties of BCC V, Ta, Mo, and W as demonstrated in [Phys. Rev. B 107, 174309 (2023)]. Similarly, the comparison of the calculated diffusivities on the homologous temperature scale shows a good agreement with experiment (Figure 5(a) of the manuscript; the remaining difference is relatively small) and in particular a very good agreement with the experimental non-Arrhenius slope (Figure 5(b) of the manuscript). Note that this is a common practice to compare the diffusion properties of different materials with the same crystalline structure. In this respect, the natural W system and the "DFT W" system are two different materials for the reasons explained above (again, the exchange potential used in DFT allows a very reasonable reproduction of the W properties, but still not *exact* at the moment) and the homologous temperature scale is the straightforward and most reliable way to compare their diffusion properties.

The quasiharmonic approximation, in a clear contrast, leads to significant errors in both the

numerical values and the general trends of vacancy formation and migration energies due to the full ignorance of the anharmonicity (Figure 2 of the manuscript). As a consequence, the non-Arrhenius behavior, i.e., the experimental non-linear slope (Figure 5(b) of the manuscript), cannot be predicted by the quasiharmonic approximation. The seemingly smaller discrepancy of the numerical values at some temperatures on the absolute temperature scale (Figure S5 in the revised supplementary material) is only coincidental due to the error compensation.

We have also explained the reasons for choosing the GGA-PBE XC in the supplementary material (Sec. IA). With the proposed computational framework and the new methodology, we have reached the accuracy limit of the selected DFT XC functional. Other available XC functionals can be straightforwardly applied with our framework in future studies. Correcting the inherent error in the DFT XC, challenging and requiring tremendous effort from the whole *ab initio* community, clearly goes beyond the scope of the present work.

Changes:

- In the revised manuscript:

Change #1: Page 7, a new discussion text has been added in the footnote: “[...] **The Arrhenius plot on the inverse homologous temperature scale compensates the significant discrepancy (more than 300 K) between the DFT (GGA-PBE) predicted melting point and the experimental melting point for BCC W, as discussed in Ref. [63]. [...]**”

Change #2: **Two new references [61] [Phys. Rev. B 109, 094110] and [63] [Phys. Rev. B 107, 174309] have been added.**

Thermal expansion

Comment 1.2: A related question to 1.1, thermal expansion can serve as another manifestation or source of anharmonicity, as suggested in e.g. Phys. Rev. Materials 6, 063402 (2022) where by accounting for the thermal expansion alone a great part of the non-Arrhenius diffusion is well captured. The authors only referred to Ref.[39] which is not the same system. Can the authors evaluate the thermal expansion coefficients obtained from their method and compare them with existing data?

Reply: We thank you for finding the incorrect citation of Ref. [39]. In the revised manuscript, we have corrected this citation, and we have additionally plotted the thermal expansion curve including a comparison with experimental data in both the Supplementary Material (Figure S1) and in this response letter (Figure R1 below).

Based on our results, the thermal expansion can be ruled out as a dominant contribution to the non-Arrhenius diffusion of W that we observe in the high temperature window up to the theoretical DFT melting point (from 1700 K to 3400 K). In both the quasiharmonic and the full vibrational calculations, the thermal expansion effect has been fully taken into account through Eq. (10) (repeated here),

$$D(T) = a_0^2(T) f_{\text{bcc}} \frac{k_{\text{B}} T}{h} \exp\left(-\frac{G_{\text{form}}(T) + G_{\text{mig}}(T)}{k_{\text{B}} T}\right), \quad (\text{R1})$$

using the equilibrium lattice constant $a_0(T)$ at the target temperatures shown in Figure R1(a) by the dots. From Figure 5(b) of the manuscript, it is apparent that the quasiharmonic results including the thermal expansion effect show minor (negligible) curvature in the Arrhenius plot, even if a wide temperature range is considered. This demonstrates the minor role of thermal expansion in explaining the non-Arrhenius diffusion of W. In contrast, the curvature of the full vibrational curve including anharmonicity aligns well with experiment. It is clear that even minor changes of G_{mig} become important due to the exponential expression, while the lattice constant squared enters the above equation. From Fig. 5 of the main manuscript, about an order

of magnitude enhancement of the W self-diffusion coefficient can be anticipated due to the non-linearity, i.e. the difference between the linear extrapolation of the low-temperature data and the measured tracer diffusion coefficients near the melting point, while the thermal expansion, Figure R1, will account only for about 5% of the resulting increment of D .

The study in [Phys. Rev. Materials 6, 063402 (2022)] was performed based on the quasiharmonic theory in a much narrower temperature window of 100–500 K and far below the melting point. The derived vacancy formation and migration enthalpies and entropies turned out to be almost temperature independent (see Fig. 5 in [Phys. Rev. Materials 6, 063402 (2022)], replotted below as Figure R2). Therefore, the non-Arrhenius feature observed within a narrow, low temperature window cannot explain the non-Arrhenius behavior at high temperatures.

Figure R1: (a) Temperature-dependent lattice constants used in this work. (b) Relative lattice expansion compared with experiments: White and Minges [Int. J. Thermophys. 18, 1269 (1997)]; Dubrovinsky and Saxena [Phys. Chem. Miner. 24, 547 (1997)].

Changes:

- In the revised manuscript:

Change #1: Correction of Ref. [41]: Phys. Rev. B 105, 045403

Change #2: Page 4: [...] The lattice constant at zero pressure $a_0(T)$ is obtained from the relative thermal expansion (DFT molecular dynamics) reported in Ref. [41] rescaled with the lattice constant at 0 K, 3.1723 Å, calculated here (see Supplementary Material for more details).

Change #3: A new reference [67] [Phys. Rev. Materials 6, 063402 (2022)] has been added.

FIG. 5. Formation enthalpy ΔH_f , migration enthalpy ΔH_m , formation entropy ΔS_f , and diffusion attempt frequency ν^* for Mo [(a), (b), (c)] and β -Ti [(d), (e), (f)], calculated with DFT within quasiharmonic transition state theory.

Figure R2: Reprinted figure with permission from [Yaxian Wang, Zhangqi Chen, Wolfgang Windl, and Ji-Cheng Zhao, *Physical Review Materials*, 6, 063402 (2022).] Copyright (2022) by the American Physical Society.

Change #4: Page 7: A new discussion text has been added: “[...] As the present DFT quasiharmonic results take the thermal expansion effect fully into account, the lattice expansion, previously proposed to explain the non-Arrhenius behavior⁶⁷, plays a negligible role in the high-temperature non-Arrhenius diffusion of W. [...]”

- In the revised Supplementary Material:

Change #1: A new “Figure S1” has been added (Page 1)

Change #2: A new discussion paragraph has been added: “Explicit calculations for $G_{\text{form}}(T)$ and $G_{\text{mig}}(T)$ were performed at eight temperatures starting at zero K and going to above the PBE melting temperature of 3349 K [3] (0 K, 600 K, 1200 K, 1700 K, 2100 K, 2500 K, 3000 K, and 3400 K), followed by a fourth order polynomial fit. For all calculations, the thermal lattice expansion effect was fully taken into account. The corresponding utilized equilibrium lattice constants are marked in Figure S1(a) on the thermal expansion curves as red (DFT full) and gray (DFT qh) dots. The equilibrium 0 K lattice constant obtained in this work is 3.1723 Å, slightly larger than the experimental value of 3.16 Å. The relative lattice expansion is displayed in Figure S1(b) in comparison to experimental data.”

Vacancies in dynamically unstable phases

Comment 1.3: A significant aspect of the outlook lies in extending this method to metastable and unstable phases with DFT accuracy. The DFT-based calculations usually suffer from not only the imaginary phonon frequencies, but also the intractability of relaxing the vacancy structure in e.g. high-temperature bcc phases of Ti and Zr, etc. Can the authors elaborate on how their method will do in such systems?

Reply: We agree with you that for dynamically unstable systems such as BCC Ti and BCC Zr the relaxation of structures with a single vacancy and also the NEB method for determining the saddle-point structures become infeasible due to the instability. For standard DFT-based methods, this is a significant challenge. The here proposed computational framework which combines thermodynamic integration with direct upsampling, provides, in contrast, a unique advantage to treat dynamically unstable phases if these phases can be stabilized in MD at high temperatures. Once a proper reference potential with known free energy is available, thermodynamic integration from the reference potential to the full vibrational potential (MTP in this work) can be applied to obtain the anharmonic free energy contribution, followed by direct upsampling to achieve the full DFT accuracy. Moreover, the final free energy is independent of the selection of the reference potential. In general, a system can be approximated by various (quasi)harmonic references which lead to different ways of splitting the vibrational free energy. Nevertheless, the sum of (quasi)harmonic and anharmonic contributions always provides the same “total vibrational free energy”, coming from atomic vibrations.

We provide in the following some ideas of how proper reference potentials could be constructed for the case of vacancy migration in dynamically stabilized phases. Further investigations are necessary to elucidate the best approach for such a purpose.

- **Vacancy-containing structures (end-points of diffusion)** Within the dynamically stabilized high-temperature regime, an effective harmonic potential fitted to DFT forces of MD snapshots or an Einstein solid (constant vibrational frequency) can be used as the reference potential. This approach has been successfully applied to study vacancy formation properties in dynamically unstable BCC Ti, Zr, and Hf (see [Phys. Rev. B 108, 184107 (2023)]).
- **Diffusion saddle point structure** The saddle point configuration can be obtained following the relaxation strategy described in [Phys. Rev. Materials 6, 063402 (2022)], i.e., interpolation of atomic positions from saddle point structures of stable BCC elements. Once the saddle point structure is obtained, the large-displacement method, also detailed in [Phys. Rev. Materials 6, 063402 (2022)], can be applied to obtain the dynamical matrix as the reference potential. Then the method for stabilization of the dynamical matrix (Eq. (4)–(6) in the main text, repeated here)

$$\underline{\underline{\mathcal{D}_s}} = \underbrace{\underline{\underline{\mathbf{w}}} \cdot \underline{\underline{\omega^2}} \cdot \underline{\underline{\mathbf{w}}}^{-1}}_{3N-1 \text{ stable modes}} + \underbrace{\underline{\underline{\mathbf{w}}} \cdot \underline{\underline{\omega_s^2}} \cdot \underline{\underline{\mathbf{w}}}^{-1}}_{1 \text{ stabilized mode}}, \quad (\text{R2})$$

where

$$\underline{\underline{\omega^2}} = \text{diag}(\omega_1^2, \dots, \omega_{3N-1}^2, 0), \quad (\text{R3})$$

$$\underline{\underline{\omega_s^2}} = \text{diag}(0, \dots, 0, \alpha|\omega_{\text{imag}}^2|) \quad (\text{R4})$$

can be applied to perform the transition state thermodynamic integration to obtain the anharmonic free energy.

Still these ideas are speculative and have to be thoroughly elaborated. Following your suggestion, we added a short discussion in the revised manuscript.

Changes:

- In the revised manuscript:

Change #1: Page 8: a new discussion has been added: “[...] **In particular for dynamically-unstable phases where the relaxation of vacancy-containing structures and also the nudged-elastic-band method become infeasible due to the instability, the present computational framework which combines TSTI with direct upsampling provides a unique advantage to treat such phases if they can be dynamically stabilized in molecular dynamics at elevated**

temperatures. Once a proper reference potential with known free energy is available, TSTI from the reference potential to the full vibrational potential (moment tensor potential (MTP) in this work) can be applied to obtain the anharmonic free energy contribution, followed by direct upsampling to achieve the full density-functional-theory accuracy. Possible references are, for example, the effective harmonic potential or the Einstein solid for the stable end-points of diffusion⁷⁵ and a migration path interpolated from stable reference elements for the transition state⁶⁷.”

Change #2: A new reference [75] [Phys. Rev. B 108, 184107] has been added.

On the electronic excitations

The authors compared the eDOS at 3000K for different structures, and revealed its strong impact on the vacancy formation and migration energies. Albeit the data plotted, the physical connection appears weak.

Comment 2.1: What is the lattice temperature during the MD simulation? How large is the mean deviation of the eDOS of the 120 MD snapshots, specifically in comparison with the difference between bulk/vacancy/transition structures? Does the DOS of vacancy and bulk structures have distinguishable differences (e.g. in [e.g. Phys. Rev. B 57, 11184 (1998)]) at 0 K and 3000 K?

Reply: The lattice temperature and the electronic temperature are the same for the eDOS calculation (specifically, 3000 K). The mean deviations of the eDOS’ (plotted in Figure R3(a)) show larger values at the valleys and at the peaks. An even more representative quantity to measure the uncertainty of the mean DOS is the standard error which is plotted in Figure R3(b). While the general features are very similar as for the mean deviation, the values are about one order of magnitude smaller, due to the high number of sampled snapshots. We also plot in Figure R3(c) a comparison between the difference in the eDOS’ of the different structures, i.e., $|\text{eDOS}_{\text{vac}} - \text{eDOS}_{\text{bulk}}|$ (black curve) and $|\text{eDOS}_{\text{ts}} - \text{eDOS}_{\text{vac}}|$ (blue curve), and the mean standard errors (red curve). It is apparent from Figure R3(c) that the standard errors are much (one order of magnitude) smaller than the mean differences between the structures in particular at the valley on the left side of the Fermi level (cf. Figure 3 of the main text), where, in contrast, the eDOS difference shows a strong peak. This part of the eDOS is the most relevant for the electronic free energy contribution. Hence, we can conclude that the mean eDOS’ averaged from 120 MD snapshots shown in Figure 3 are well converged.

A comparison of the eDOS’ at 0 K and 3000 K for the bulk and vacancy structures is plotted in Figure R4. At 0 K, the eDOS profiles (dashed lines) are quite similar except for minor shifts in the peaks and valleys. In particular, we do not observe the significant peak for the vacancy structure at the valley close to the Fermi level as shown in [Phys. Rev. B 57, 11184 (1998)]. The eDOS’ calculated in the 1998 paper are from very small supercells of 27 atoms which are neither sufficient to converge the vacancy energetics nor the eDOS. In the present work, we utilize larger supercells with 128/127 atoms (bulk/vacancy) and we mainly focus on the high-temperature eDOS’ (solid lines in Figure R4) which exhibit a significant smearing effect, as compared to the peaks and valleys at 0 K. The eDOS of the vacancy structure (red solid line) is on average slightly higher than that of the bulk structure (blue solid line) at the valley close to the Fermi level, resulting in a negative free energy contribution to the vacancy formation energy as discussed in the main text.

Changes:

- In the revised manuscript:

Change #1: On Page 5, a short discussion about the convergence of the eDOS has been added: “[...] **Despite the considerable variance in the eDOS’s of the single snapshots,**

Figure R3: A statistical analysis of the eDOS': (a) mean deviations, (b) standard errors and (c) a comparison between the eDOS differences and the mean standard error.

Figure R4: Electronic DOS' at 0 K and 3000 K for bulk and vacancy structures.

the standard errors of the mean values are about one order of magnitude smaller than the differences between the different types of structures (bulk, vacancy, and transition state) in particular at the valleys and peaks (see Figure S4 in Supplementary Material), suggesting that the eDOS's are well converged [...]"

- In the Supplementary Material:

Change #1: A new section “III ELECTRONIC DOS CONVERGENCE” has been added with the discussion text: “To quantify the uncertainty of the mean eDOS's shown in Figure 4 of the main text, we plot the standard errors in Figure S4(a) and compare them with the differences between different types of structures, i.e., $|eDOS_{\text{bulk}} - eDOS_{\text{vac}}|$ and $|eDOS_{\text{vac}} - eDOS_{\text{ts}}|$, in Figure S4(b). While the general features in terms of peaks and valleys between the standard error and the differences are similar, the values are about one order of magnitude smaller in particular at the valleys and peaks of the eDOS's where the differences shows a peak. This means the mean eDOS's averaged over 120 MD snapshots shown in Figure 4 of the main text are well converged.”

Change #2: A new Figure S4 has been added.

Comment 2.2: In Fig. 2b, the migration energy for the “qh+ah+el” data point seems to be larger than that of the “qh+ah”. Can the authors comment on this? The underlying physical connection between the electronic temperature (if it is simply the smearing factor in the MD calculations) and the vacancy formation/migration energies is not very clear.

Reply: The electronic free energy contribution to the vacancy migration energy is negative and increases in magnitude with increasing temperature. This means that at high temperatures the migration energies for “qh+ah+el” (red curve) are all smaller than the ones for “qh+ah” (blue curve). At low temperatures, e.g., below 1000 K, the electronic contribution is very small and close to zero. The very small difference between the blue and red point at 600 K is only about 0.003 eV (0.02 meV/atom) which is already at the limit of the averaging statistics since electronic free energies are calculated from the thermal average of MD snapshots.

As discussed in detail in [Phys. Rev. B 95, 165126 (2017)], the physical connection between the electronic temperature and the electronic free energy can be understood from the

corresponding free energy expressions:

$$F^{\text{el}}(T) = U^{\text{el}}(T) - TS^{\text{el}}(T), \quad (\text{R5})$$

$$U^{\text{el}}(T) = \int_{-\infty}^{\infty} D(\varepsilon) f \varepsilon \, d\varepsilon - \int_{-\infty}^{\varepsilon_{\text{F}}} D(\varepsilon) \varepsilon \, d\varepsilon, \quad (\text{R6})$$

$$S^{\text{el}}(T) = 2k_B \int_{-\infty}^{\infty} D(\varepsilon) s(\varepsilon, T) \, d\varepsilon, \quad (\text{R7})$$

where

$$s(\varepsilon, T) = -[f \ln f + (1 - f) \ln(1 - f)] \quad (\text{R8})$$

and the Fermi function $f = f(\varepsilon, T) = \left[\exp\left(\frac{\varepsilon_i - \varepsilon_{\text{F}}}{k_B T}\right) + 1 \right]^{-1}$, where ε_i are the electronic energies of the Kohn-Sham eigenstates, ε_{F} is the Fermi energy or level, and k_B the Boltzmann constant. As seen from the equations, the electronic free energy F^{el} is directly linked to the electronic DOS $D(\varepsilon)$. The latter is, however, modified by the Fermi function $f(\varepsilon, T)$ at the respective (electronic) temperature and therefore only the DOS close to the Fermi level contributes to F^{el} . For the internal energy, this can be understood by realizing that the first integral in Eq. (R6) is canceled by the second one whenever $f \approx 1$, i.e., at energies sufficiently below the Fermi level. For energies sufficiently above the Fermi level, i.e., $f \approx 0$, both integrals give a negligible contribution. Regarding the entropy, the entropy function $s(\varepsilon, T)$ in Eq. (R7) is only peaked around the Fermi level as illustrated in Figure R5. As the (electronic) temperature increases, the width of the peak gets moderately larger (broadening effect, see Figure R5) but the contribution of the eDOS to the electronic free energy still stays confined to about ± 1 eV around the Fermi level. Therefore, the Fermi smearing effect only plays a critical role in the electronic free energy in this region around the Fermi level, re-populating the energy states below the Fermi level to higher energy states above the Fermi level, as illustrated in the right panel of Fig. 3 in the main text. Consequently, a *large* DOS at the Fermi level implies a *largely negative* electronic free energy. Accordingly, the vacancy (transition state) structure has a more negative electronic free energy than the bulk (vacancy) structure, i.e., a negative free energy contribution to the formation (migration) energy.

Figure R5: Entropy function $s(\varepsilon, T)$.

Changes:

- In the revised manuscript:

Change #1: A new reference [56] [Phy. Rev. B 95, 165126 (2017)] has been added.

Change #2: Page 5: a new discussion has been added: “[...] At high (electronic) temperatures, the Fermi function (dash dotted curve shown in the right panel) re-populates the electrons from below the Fermi level to higher energy states above the Fermi level. The resulting holes and excited electrons are the origin of the electronic entropy contribution to the free energy. A *large* eDOS at the Fermi level facilitates many such excitations and thus implies a *largely negative* electronic free energy, as detailed in Ref. [56]. As observed around the valley region enlarged in the right panel of Figure 4, the eDOS’s close to the Fermi level increase with the sequence of bulk to vacancy to transition state. [...]”

On the collective migration

The projection of the atomic trajectories onto different migration modes provides an interesting perspective regarding a general understanding of anomalous self-diffusion behaviors in many systems.

Comment 3.1: In the long-time molecular dynamics simulation specifically, how is the lattice degree of freedom controlled? What is the major difference in performing calculations with “full” vibrations vs. “harmonic” vibrations?

Reply: The lattice degree of freedom is controlled by the corresponding forces generated either by the harmonic dynamical matrix (harmonic vibrations) or the moment tensor potential (full vibrations). Specifically for the transition state, we utilize a stabilized dynamical matrix (Eq. (4) in the main text; Eq. (R2) above) to generate harmonic forces and the MTP forces described by Eq. (8) (repeated here),

$$\mathcal{F}_s^{\text{MTP}} = \mathcal{F}^{\text{MTP}}(\{\mathbf{R}_I\}) + [\mathcal{F}_s^{\text{qh}}(\{\mathbf{R}_I^{\text{sp}} + \mathbf{u}'_I\}) - \mathcal{F}^{\text{MTP}}(\{\mathbf{R}_I^{\text{sp}} + \mathbf{u}'_I\})], \quad (\text{R9})$$

stabilized along the unstable mode for the full vibrational simulations.

Comment 3.2: What is the color coding in Fig.4(b-e)? Is it corresponding to the time evolution, or is it some kind of histogram? Other than the projected intensity on different migration vectors, can the authors show some real-time snapshots like for example in Refs. [Nat. Geosci. 10, 312 (2017), Phys. Rev. Lett. 123, 105501 (2019)] to illustrate the collective diffusion?

Reply: The color coding in Fig. 4(b)–(e) [(c)–(f) in the revised manuscript] represents the distribution density of the vibrational phase space (the same as for a distribution histogram). In the revised manuscript, we have added a color-bar in Fig. 4 of the main text to indicate the meaning of the color coding.

Regarding the real-time trajectory, we have added a new sub-figure in Figure 4 showing the real-time MD trajectory (the white line associated with each atom) of a short timeframe of consecutive 300 MD steps ($300 \times 5 \text{ fs} = 1.5 \text{ ps}$). The collective motion of the second neighbor atom (green) and the migrating atom (blue) can be clearly observed (highlighted by yellow arrows).

Changes:

- In the revised manuscript:

Change #1: A color-bar has been added in Fig. 5 of the main text.

Change #2: A new sub-figure (b) illustrating the real-time trajectory has been added to Figure 5.

Change #3: The caption of Figure 5 has been modified: “[...] (b) Real-time MD trajectory (the white line associated with each atom) of a short timeframe of consecutive 300 MD steps (yellow arrows highlight the directions of the collective motion). [...] and the corresponding index of each sub-figure has been updated.”

Change #4: Page 7: The related discussion text has been modified: “With the presence of the migrating atom at the saddle point, most interestingly and not reported so far, such

an anharmonic inward motion of the 2nd neighbor atom is significantly correlated with the motion of the migrating atom: it is particularly favored when the migrating atom moves away from the vacancy (toward the other vacant site), as is shown by the non-symmetric distributions in quadrant I and II of Figure 5(d) and highlighted by the yellow arrows in the real-time MD trajectory depicted in Figure 5(b).”

Miscellaneous

I found inconsistency in e.g. figure references between the SI and the main text and suggest the authors to give a careful check.

Reply: We thank you for the very careful check. In the revised manuscript and Supplementary Material, we have corrected these inconsistencies.

Changes:

- In the revised Supplementary Material:

Two places referenced with “Figure 3” of the main text have been changed to “Figure 4”.

Responses to Reviewer #2

General comment: In many pure metals, it has been observed that the measured diffusivity due to vacancy-atom exchanges deviates from Arrhenius behavior. More specifically, experiments are rationalized by assuming that the activation energy, Q , and the diffusivity prefactor D_0 are constants, independent of temperature.

Deviations from Arrhenius behavior have been typically ascribed to the presence of defects other than mono-vacancies as the mediators for atomic substitutional exchange. Di-vacancies, for example, have been invoked to explain deviations from Arrhenius behavior. The authors point out that other explanations have been invoked, including contributions due to anharmonicity. In this manuscript the authors address the problem using a highly novel and efficient computational method to predict the different factors that, together, control the vacancy-mediated diffusion in metals. They proceed to apply the framework to investigate the supposed diffusion anomaly in Tungsten and further demonstrate the utility of the framework to predict the diffusivity in hcp-based high entropy alloys.

On the methodological front, this paper is impeccable. The authors clearly explain the limitations of current DFT-based approaches to predict the thermodynamics of non-equilibrium states, such as those corresponding to a diffusing atom 'sitting' a top of an energy barrier as it transitions from one substitutional site to another through exchanging places with a vacancy. To address this issue the authors employ a state-of-the-art machine learning potential to attain energetic predictions at the DFT-level of precision. They then use highly advanced thermodynamic integration tools to estimate the temperature dependence of the vacancy formation and migration Gibbs free energies, accounting for quasi-harmonic, electronic and anharmonic effects. The anharmonicity in this case is assumed to arise (at least in the case of the migration Gibbs energy) from the non-harmonic shape of the crystal field along the direction of atomic migration.

While the method used represents the state of the art and the culmination of the trajectory towards increased fidelity in the estimation of thermodynamic properties by this group (among the leaders in the field), the results and interpretation fall short. Below some of the most salient issues are presented:

Reply: We thank you for the very positive evaluation of our work as well as the constructive comments and suggestions for improving the manuscript. We hope we have addressed your concerns with the following responses and revisions.

On the di-vacancy hypothesis

Comment 1: Perhaps the most evident problem is the fact that the authors are using an advanced theoretical approach to test the hypothesis that non-Arrhenius behavior in the diffusion of W is due to anharmonic contributions to the vacancy's migration and formation free energies. However, this test does not consider the commonly invoked alternative hypothesis, namely, that di-vacancies may be responsible for (at least some of) this deviation from Arrhenius behavior. Trying to validate a theory at the DFT level with experiments is very challenging (more on this later), so a first step would be to carry out an analysis assuming that di-vacancies are present and then try to come up with arguments based on theoretical grounds and some type of comparison with experiments in order to establish the validity of either of the two competing hypotheses (there could be more explanations).

Reply: First, we would like to highlight that the present paper is dedicated to elaboration of a general approach to address the anharmonicity of the transition states. In both cases, single vacancy jump or di-vacancy migration, the free energy of migration has to be evaluated and the present paper provides a framework to address the problem.

We agree with you that we have not explicitly considered the hypothesis of the di-vacancy mechanism which, in principle, may contribute to the high-temperature non-Arrhenius diffusion

of BCC W. However, we do not believe so due to the following reasons.

In a previous high-accuracy DFT study (Ref. [23] of the main text, [Phys. Rev. X 4, 011018 (2014)]), the di-vacancy mechanism has been convincingly ruled out in FCC elements (i.e., Al and Cu) by explicit finite-temperature DFT calculations of di-vacancy formation Gibbs energies. Therein, clear contradictions with experimental observations in terms of the vacancy concentration and the formation entropy were obtained for the di-vacancy hypothesis. For BCC refractory elements like tungsten, due to the significant challenges in experimental measurements of mono- and di-vacancy concentrations at very high temperatures, the same set of arguments as given in Ref. [23] for FCC elements cannot be directly utilized. We thus provide here (and in the revised Supplementary Material) an alternative analysis based on experimental diffusion data and the available monovacancy and di-vacancy energetics to substantiate that di-vacancies can hardly explain the non-Arrhenius diffusion of BCC W.

By assuming a collective mechanism of diffusion (i.e., mono- and di-vacancies) in BCC W, the diffusivity can be expressed as:

$$D = \frac{c_{1v}}{c_{1v} + 2c_{2v}} D_{1v} + \frac{2c_{2v}}{c_{1v} + 2c_{2v}} D_{2v} \quad (\text{R10})$$

$$= f_1 D_{1v,0} \exp\left(-\frac{Q_{1v}}{k_B T}\right) + f_2 D_{2v,0} \exp\left(-\frac{Q_{2v}}{k_B T}\right), \quad (\text{R11})$$

where c_i and D_i ($i = 1v$ or $2v$) refer to the concentration of monovacancies ($1v$) or di-vacancies ($2v$) and their diffusivity contributions, respectively. Equation (R10) can be rewritten as shown in Eq. (R11), i.e., in terms of the fraction f_i of each type i of vacancy, the pre-factor $D_{i,0}$, and the activation energy Q_i . Fitting the measured diffusivities to Eq. (R11), as shown in Ref. [12] of the main text [Mundy 1978], results in:

$$D = 0.04 \exp\left(-\frac{5.45 \text{ eV}}{k_B T}\right) + 46 \exp\left(-\frac{6.9 \text{ eV}}{k_B T}\right), \quad (\text{R12})$$

where the first term would correspond to the mono-vacancy diffusion since the activation energy (5.45 eV) agrees well with the sum of the mean values of the available experimental data for the mono-vacancy formation energy (3.8 eV) and migration energy (1.7 eV). It must be then that the second term would correspond to di-vacancy diffusion; however, in such a case a clear contradiction can be derived as follows.

The activation energy of the di-vacancy diffusion Q_{2v} can be written in terms of the monovacancy formation energy H_{form}^{1v} , the binding energy of a vacancy pair H_{bind}^{2v} , and the di-vacancy migration energy H_{mig}^{2v} as

$$Q_{2v} = 2H_{\text{form}}^{1v} - H_{\text{bind}}^{2v} + H_{\text{mig}}^{2v}. \quad (\text{R13})$$

The di-vacancy binding energy then reads

$$H_{\text{bind}}^{2v} = 2H_{\text{form}}^{1v} + H_{\text{mig}}^{2v} - Q_{2v}. \quad (\text{R14})$$

The experimental di-vacancy migration energy was reported by Park et al. (Ref. [47] of the main text; [Philos. Mag. A 48, 397–419]) and by Rasch et al. (Ref. [48] of the main text; [Philos. Mag. A 41, 91–117]) to be very similar to the mono-vacancy migration energy, i.e., about 1.8 eV. This was also confirmed by a recent DFT-based kinetic Monte Carlo study in [Nucl. Fusion 58 (2018) 026004] where an effective di-vacancy migration energy of 1.65 eV was obtained. Based on these experimental mono-/di-vacancy energies together with the di-vacancy diffusion activation energy reported by Mundy et al., the di-vacancy binding energy can be calculated as

$$\begin{aligned} H_{\text{bind}}^{2v} &= 2H_{\text{form}}^{1v} + H_{\text{mig}}^{2v} - Q_{2v} \\ &= 2 \times 3.8 \text{ eV} + 1.8 \text{ eV} - 6.9 \text{ eV} = 2.5 \text{ eV}. \end{aligned} \quad (\text{R15})$$

Now, the value of $H_{\text{bind}}^{2v} = 2.5 \text{ eV}$ is, however, dramatically larger than directly obtained theoretical and experimental values: DFT studies have shown very small di-vacancy binding energies

ranging from -0.1 eV to $+0.05$ eV (see [Nucl. Fusion 58 (2018) 026004]). With our own DFT calculations, we also obtained a small 0 K binding energy of -0.01 eV. The only available experimental value is 0.7 eV reported by Park et al. (Ref. [47] of the main text; [Philos. Mag. A 48, 397–419]), which is still well below 1 eV. Such a large discrepancy with respect to the value in Eq. (R15) cannot be explained by uncertainty. Therefore, the di-vacancy diffusion cannot be used to explain the non-Arrhenius diffusion of BCC W, or at least it must be that it only plays a minor role.

In fact, the consistent curvature of the full DFT results with experiment obtained in the present study based on a single mono-vacancy mechanism is clear and convincing evidence that the di-vacancy mechanism contribution is not appropriate in explaining the non-Arrhenius dependence.

Changes:

- In the revised manuscript:

Change #1: Page 7: A short discussion text has been added: “Since the calculations are based on the single mono-vacancy mechanism, the consistent curvature of the full DFT results with experiment indicates that the alternative di-vacancy mechanism plays a minor role in the non-Arrhenius diffusion of tungsten. A detailed analysis on the di-vacancy hypothesis, likewise leading to an exclusion of this mechanism, is provided in Supplementary Material.”

- In the revised Supplementary Material:

Change #1: A new section “VI ON THE DI-VACANCY HYPOTHESIS” has been added based on the reply above to provide a detailed discussion of the di-vacancy hypothesis also to the readers.

Arrhenius extrapolation

Comment 2: Another problem has to do with the way the authors try to establish the correctness of their predictions by comparing against a sparse set of experimental results. The authors seem to extrapolate (Fig 2) the values of the high-temperature migration and formation energies to low temperature, claiming that this is analogous to approaches whereby experimental temperature series have been extrapolated to low temperatures. A first question is then why not comparing the measurements done at high temperatures with the predictions carried out at similar temperatures? It is not clear why the authors chose to only compare extrapolations to low temperatures when they produce high-temperature predictions and experiments (as claimed by these authors) have been carried out at similarly high temperatures.

Reply: Experimental measurements of vacancy formation and migration energies, either by positron annihilation spectroscopy or resistivity measurements, are inevitably performed at high temperatures (the temperature intervals of the measurements are explicitly indicated in the corresponding plot, see e.g. Figure R6 below). The measured raw data, i.e., the life time data or the resistivity data, are related to the vacancy formation/migration *Gibbs energy*. However, in order to experimentally extract vacancy properties or even diffusion parameters, an Arrhenius ansatz, i.e., a linear temperature dependence of the Gibbs energy (which is not correct) instead of the true temperature dependence as revealed in this work, is generally applied in the fitting procedures as well as in various models, i.e.,

$$G_{\text{form/mig}} = H_{\text{form/mig}} - TS_{\text{form/mig}} \quad (\text{R16})$$

where $H_{\text{form/mig}}$ and $S_{\text{form/mig}}$ are *temperature-independent* enthalpy and entropy of formation to be fitted and reported in the literature. Applying such an Arrhenius fit, as illustrated by the dashed line “Arrhenius extrapolation” in Figure 2 of the manuscript, results in notable

discrepancies in the obtained formation/migration enthalpy compared to the values computed by the standard 0 K DFT values. The breakdown of the experimental Arrhenius extrapolation has been discussed in detail in Ref. [23] of the main text [Phys. Rev. X 4, 011018 (2014)].

If we are to compare with the linearly extrapolated experimental values, we have to apply the same linear extrapolation to our explicitly computed high temperature data. Although such a linear extrapolation yields *per se* incorrect results as just discussed, it still makes sense to explicitly analyze it due to its universal usage in the community. Our explicit analysis should make the shortcoming evident.

Changes:

- In the revised manuscript:

Change #1: Page 5: text modified: “The experimental data (black symbols) reflect extrapolations of high temperature measurements utilizing the *linear* Arrhenius ansatz, i.e., $G_{\text{form/mig}} = H_{\text{form/mig}} - TS_{\text{form/mig}}$, with temperature-independent enthalpies ($H_{\text{form/mig}}$) and entropies ($S_{\text{form/mig}}$) of formation/migration. Although such a linear fitting ansatz is clearly inappropriate based on the present and previous results^{23,24}, we utilize it nevertheless to fit our exact data for the purpose of an unbiased comparison between theory and experiment. For both formation and migration, a linear Arrhenius fit of the full Gibbs energies between 2400 K and 2800 K shows good agreement with experiment considering the significant scatter and uncertainty in the high temperature measurements. ”

Comment 3: Another question that arises when examining their analysis in Fig. 2. Specifically, they choose to extrapolate to low temperatures their calculations considering anharmonic effects. Yet, at least in the case of the vacancy migration energy, it seems that extrapolating from the ‘qh+el’ approximation may also yield results that would be comparable to experimentally measured values, within the uncertainty of the experiments.

Reply: The vacancy migration Gibbs energies from the “qh+el” approximation are plotted in Figure R6 (blue curve). This curve includes the contribution due to the electronic free energy and the coupling effect between full vibrations and electronic excitations. The Arrhenius extrapolation (red dashed line) to low temperatures shows a significant discrepancy with respect to the relevant high-temperature experimental data (triangle with error bar corresponding to the measurements at 2600 K).

Figure R6: Migration Gibbs energies predicted by the “qh+el” approximation.

Uncertainty analysis

Comment 4: This last point brings another question: the authors are presenting their computational predictions in a deterministic manner and no uncertainty bounds are presented in their calculations. This is definitely not correct as surely there must be a number of factors that would contribute to uncertainty in the predictions, including calculations setups, pseudopotential choice, uncertainties arising from numerical uncertainties in their thermodynamic integration, uncertainties due to imperfect training of the ML potential, etc. Over the past decade there have been a number of works that have pointed out the importance that accounting for uncertainty in computational materials science predictions if one is to compare with experiments.

Reply: We agree with the referee that the uncertainty estimation of the calculated data has been missing in the original manuscript. Therefore, we have added in the revised Supplementary Material a new section “UNCERTAINTY ANALYSIS” where we carefully estimate/calculate the statistical uncertainties based on standard statistical approaches. The detailed uncertainty values are given in Table S1 in the Supplementary Material. In general, the calculated uncertainty values are very small compared to the actual data points of both Gibbs energies and diffusivities, which means that none of the arguments and conclusions are affected by these small uncertainties.

Other factors that may introduce uncertainties are also discussed in detail in the significantly extended “Section I TECHNICAL DETAILS” of the Supplementary Material. In particular, all relevant calculation setups, i.e., DFT calculations, nudged elastic band calculations, thermodynamic integration, and direct upsampling, are described and checked to ensure the full convergence up to DFT accuracy.

The error analysis of the machine learning potential (Page 2 of the revised manuscript) with two new subfigures in Figure 1 of the revised manuscript have been added. Nevertheless, we need to stress the accuracy of the final results is not affected by the error in the machine learning potential (i.e., the MTP accuracy) because the direct upsampling calculations bring the final accuracy to the DFT level.

Changes:

- In the revised Supplementary Material

Change #1: Section I “TECHNICAL DETAILS” has been significantly extended where all calculation setups are discussed and checked to ensure the full convergence.

Change #2: A new section “UNCERTAINTY ANALYSIS” has been added with a detailed estimation of the uncertainty of the calculated data.

Change #3: A new table (Table S1) has been added for a compilation of the uncertainty values.

Non-Arrhenius behavior

Comment 5: One last problematic aspect of the paper has to do with Fig. 5. In Fig. 5a, the authors present a number of temperature series corresponding to their own calculations under different levels of approximation as well as some experimental data. From their results, it is evident that their predictions where they account for all the contributions suggest a non-Arrhenius behavior of the diffusivity of tungsten. The experiments presented tend to be consistent with each other with some scattering.

While there may seem that there is some deviation from linearity in $\log - 1/T$ coordinates, I believe that the analysis presented in Fig. 5b to reinforce the authors’ argument for the importance of anharmonicity in the diffusion of tungsten is not really as strong as they suggest. The biggest issue is that they seek to make their case by taking a derivative of their predictions as a function of temperature and do the same on a fit to an experimental dataset. While I accept that their own data exhibits a deviation from “0” that suggests that the prefactor and

migration energies may be temperature-dependent, the same cannot be said, unquestionably, in the case of the experimental dataset. After all, the temperature dependence of these quantities is determined by the form of the function used to fit the data. If, for example, a linear fit would have been used, then there would not be any possibility for inferring a non-Arrhenius behavior in the experimental data. In fact, while all the data is in agreement, there is sufficient scatter in the experimental data to preclude any kind of certainty in the use a specific function or other when fitting the dataset.

Reply: We agree with you that one has to be careful in selecting the form of the function for fitting the experimental data considering the scatter. It should be clarified first that the fit of the experimental data (black line) shown in Figure 5 is obtained by Mundy et al. from a single set of experimental data (black dots) and does not use data from different and unrelated experimental studies (black + color symbols). In this way, it is ensured that potential systematic errors from unrelated experiments do not distort the fitting procedure. Some unavoidable statistical experimental scatter can be observed in the dataset of Mundy et al. However, the important point is that, regardless of the scatter, there is a clear non-linear trend visible on the wide temperature scale, even without any fitting. Now, this means that in order to reproduce the trend of the measured data properly, a linear fit is *not* sufficient and higher order terms have significant, *non-negligible* contributions. Note this issue has been extensively discussed in the original work of Mundy et al. and their discussion represents the current viewpoint.

To highlight the importance of the higher-order non-linear contributions, we plot in the following (Figure R7) the non-linear contribution to the diffusivities. We first define the linear extrapolation of the low-temperature quasiharmonic diffusivities as the reference D^{ref} . We then subtract these reference values from all the data shown in Figure 5 of the main text and plot the remaining diffusivity contribution in Figure R7. It is apparent that the higher-order non-linear contributions are significant even considering the scatter. This strongly supports the unquestionable existence of the observed non-Arrhenius behavior and the validity of the fitting.

Actually the non-Arrhenius behavior in the diffusivities is already indicated by the strong curvatures of the vacancy formation and migration Gibbs energies shown Figure 2 of the main text. In general, any non-linearities in these temperature-dependent formation or migration Gibbs energies result in non-Arrhenius behavior in diffusivities.

Figure R7: Non-linear contribution to the diffusivities.

Comment 6: At the introduction of their manuscript, the authors correctly state that (paraphrasing), extraordinary claims require extraordinary evidence. Unfortunately, in this case,

while the authors do present some plausible evidence for non-Arrhenius diffusivity in tungsten, the case is by no means strong. In fact, the lack of rigorous uncertainty/statistical analysis makes it impossible to assert any level of confidence in their conclusions. It is not clear, to me at least, that the authors have made their case with 100% certainty. It is probably not the case that this certainty is 0%. It must therefore be somewhere in between but without proper analysis it is not possible to ascertain the correctness of their predictions. These deficiencies in no way detract from the excellent and much-needed methodological improvements in first principles predictions.

Reply: We thank you for the appreciation of the significance of the present work including the methodological advancement. In the revised supplementary material, with the rigorous uncertainty/statistical analysis additionally added in Sec. V and the extensive convergence details added in Sec. I, we believe that the revised manuscript and supplementary material have addressed your concern on the uncertainty of the predictions. We are confident that all presented results and findings are physically correct and reliable.

Responses to Reviewer #3

General comment: The manuscript titled "Ab initio machine-learning unveils strong anharmonicity in non-Arrhenius self-diffusion of tungsten" introduces a numerical framework integrating Machine Learning, Density Functional Theory (DFT) calculations, and canonical sampling methods. This framework adeptly calculates the free energy associated with the formation and migration of a mono-vacancy in tungsten. By directly estimating these quantities, the authors determine the temperature-dependent self-diffusion coefficient of tungsten. They conclude that the anomalous high-temperature behavior of this coefficient, deviating from the Arrhenius model, is due to anharmonic and electronic contributions in the free energy of this basic defect.

The theoretical and numerical literature on this subject has been contentious, particularly regarding the cause of the experimentally observed non-standard Arrhenius behavior of the self-diffusion coefficient at high temperatures. Many studies in the late '70s and early '80s investigated the origin of this phenomenon. Anharmonicity was proposed but integrated into the general thermodynamic model without support from atomistic calculations. In the '90s and late 2000s, the prevailing view was that the curvature in the self-diffusion coefficient stemmed from di-vacancies rather than mono-vacancies. This is where this study excels; the authors attempt to clarify this issue. The results align with experimental data trends, although they consistently overestimate the self-diffusion coefficient.

However, despite the positive aspects, I have significant reservations about this study. Although the concept of a completely coupled framework is intriguing, the central method for sampling the saddle point with ab initio accuracy seems too far-fetched and mathematically imprecise. The authors possess extensive experience in sampling minima, but it appears that saddle points (a parabolic energy surface with one unstable mode of the Hessian) fall outside their expertise. The free energy in the current framework is inevitably biased. It may veer towards either the lower or upper limit of the true value, but there is a systematic error in the evaluation of sampling averages. This critical issue is not addressed in the paper, and I consider it a central flaw that could affect the final results. Unless this issue is resolved, I cannot envision a path to publication for this paper. To put it succinctly, in its current form, I do not recommend the publication of the paper. From a mathematical standpoint, I find the results to be irrelevant.

Reply: We thank you for the informative overview of understanding the non-Arrhenius diffusion phenomenon that has been long debatable and for the appreciation of the contribution of the present work. We will address in the following your concerns in terms of the possible bias in the simulations with detailed explanations and additional data support.

Mixing of the forces

Comment 1: In this method the quasiharmonic decomposition of the Hessian is used in order to compensate 0 K Hessian unstable mode. Then by fixing the the interatomic force as a mix between the DFT quasiharmonic and MTP force fields by linear interpolation with some tricky projection in the unstable direction, the TI integration of performed in this completely parabolic as-for-a-minimum energy landscape. Here are two major sources of bias:

The first source of bias is the mixing of forces. Based on the equations presented in the section "Transition state thermodynamics integration," it is unclear how the authors carry out the sampling method. The text and equations create confusion between classical TI (equation 3) and equation 7, where forces are mixed or averaged through a mixing parameter. The authors do not clarify the sampling dynamics of the external parameter λ (in equation 7), which unfortunately has the same notation but a completely different meaning compared to equation 3.

Reply: The proposed transition state thermodynamic integration (TSTI) is based on the standard equilibrium thermodynamic integration (TI), i.e., the free energy difference between potentials A and B is calculated by defining a thermodynamic path between A and B and integrating

over the ensemble-averaged energy difference between A and B along the path. The standard TI procedure to compute the anharmonic free energy of a usual stable system is described in the following and related to TSTI.

Within standard equilibrium TI, the explicitly anharmonic free energy, i.e., the free energy difference between the full vibrational potential (i.e., MTP in the present case) and the quasi-harmonic potential is calculated along a linearly-mixed thermodynamic pathway between the quasi-harmonic potential E^{qh} and the moment tensor potential (MTP) E^{MTP} , i.e.,

$$E_\lambda = (1 - \lambda)E^{\text{qh}} + \lambda E^{\text{MTP}}, \quad (\text{R17})$$

where λ is a coupling parameter with value between 0 and 1. To perform the integration, canonical ensemble MD simulations are generally executed for a set of different λ values. For each *fixed* value of λ , the atomic movement in the MD simulations is driven by forces \mathcal{F} corresponding either to $E^{\text{qh}}(\lambda = 0)$ or $E^{\text{MTP}}(\lambda = 1)$ or a linear mixing of both $E_\lambda(0 < \lambda < 1)$:

$$\mathcal{F}_\lambda = (1 - \lambda)\mathcal{F}^{\text{qh}} + \lambda\mathcal{F}^{\text{MTP}}. \quad (\text{R18})$$

The anharmonic free energy then reads

$$F^{\text{ah}} = \int_0^1 d\lambda \left\langle \frac{\partial E_\lambda}{\partial \lambda} \right\rangle_\lambda = \int_0^1 d\lambda \langle E^{\text{MTP}}(\{\mathbf{R}_I\}) - E^{\text{qh}}(\{\mathbf{R}_I\}) \rangle_\lambda, \quad (\text{R19})$$

where $\langle \dots \rangle$ refers to the thermal average and $\{\mathbf{R}_I\}$ to the set of atomic coordinates.

Equations (7) and (3) in the main text (repeated here),

$$\text{Eq. (7) main text:} \quad \mathcal{F}_\lambda = (1 - \lambda)\mathcal{F}_s^{\text{qh}} + \lambda\mathcal{F}_s^{\text{MTP}}, \quad (\text{R20})$$

$$\text{Eq. (3) main text:} \quad \Delta F_{\text{MTP}}^{\text{qh} \rightarrow \text{full}} = \int_0^1 d\lambda \langle E^{\text{full}}(\{\mathbf{R}_I\}) - E^{\text{qh}}(\{\mathbf{R}_I\}; \underline{\mathcal{D}}_s) \rangle_\lambda \quad (\text{R21})$$

$$F_{\text{TS}}^{\text{ah}} = \int_0^1 d\lambda \langle E^{\text{MTP}}(\{\mathbf{R}_I\}) - E^{\text{qh}}(\{\mathbf{R}_I\}; \underline{\mathcal{D}}_s) \rangle_\lambda, \quad (\text{R22})$$

correspond to Eqs. (R18) and (R19) here. In Eq. (R22), we have just rewritten Eq. (R21) in terms of two new symbols, $F_{\text{TS}}^{\text{ah}}$ (TS=transition state) and E^{MTP} instead of $\Delta F_{\text{MTP}}^{\text{qh} \rightarrow \text{full}}$ and E^{full} , in order to make the correspondence more explicit. The coupling parameter λ has the same meaning in both Equations (7) and (3) as well as in Eqs. (R18) and (R19). The requirement and treatment of mixing the forces come from the standard equilibrium TI instead of the new TSTI approach. Therefore, we do not introduce any bias in the mixing of forces. The main difference between Equation (7) and Eq. (R18) is the stabilization treatment for \mathcal{F}^{qh} and \mathcal{F}^{MTP} as described in the main text.

Changes:

- In the revised manuscript:

Change #1: Page 2, additional discussion text has been added: “[...] For each *fixed* value of λ , the atomic movement in the MD simulation is driven by forces derived from a linear combination of quasi-harmonic and MTP forces.”

Migration pathway

Comment 2: The second source of bias arises from the authors’ use of a biasing quasi-harmonic method to circumvent the impossibility of direct sampling at the saddle point. This approach is based on the assumption that the flow direction at the saddle point (which is identified as the Hessian eigenvector at 0K) exhibits low anharmonicity and its direction remains unchanged vectorially at any temperature of the sampling. This hypothesis is unverified, and other studies,

even in low-dimensional and simple energy landscapes, easily demonstrate the contrary. The pathways at 0K are not the same as those at finite temperatures, and the authors do not account for this discrepancy in their sampling approach, unlike the treatment proposed by D. Branduardi, F. L. Gervasio, and M. Parrinello in "From A to B in Free Energy Space," J. Chem. Phys. 126, 054103 (2007).

The bias in the free energy calculation is not quantified, which is a significant concern. The authors' statement that "the unstable mode, which drives the saddle-point configuration to equilibrium, does not contribute to the vibrational free energy of the transition state and can be modified to a stable one without altering the free energy" demonstrates a fundamental misunderstanding of sampling methods. Although the unstable mode does not contribute to the free energy after thermodynamic integration, the sampling is skewed by the quasi-harmonic potential. This major methodological flaw raises serious doubts about the analysis of the results.

Reply: We agree that the sampling at the saddle point in the TSTI approach relies on the 0 K migration pathway as obtained from the NEB calculation. In order to verify the validity of this hypothesis at high temperatures, we performed additional MTP-MD runs at 3400 K (the highest temperature considered in this work) with which we have sampled in total 754 high-temperature migration pathways. The resulting statistical plot of all collected migration pathways is shown in Figure R8 where the gray dots indicate the trajectories of atoms at the BCC corners vibrating and migrating toward the vacancy in the middle of the BCC unit cell (cube with the black edges). It is apparent that, on average, the migration pathways of the eight nearest neighbors of the vacancy preserve the full BCC crystalline symmetry, migrating along the $\langle 111 \rangle$ direction which is the same as for the NEB path at 0 K. Moreover, we show the distribution density at the right bottom corner by cross-sections cut by the (110) plane and the (111) plane. The resulting statistical Gaussian distributions are centered along the [111] direction, thus further supporting that at high temperatures the migration pathways follow the 0 K NEB pathway within an ensemble average. This is due to the fact that for highly-symmetrical crystalline structures like BCC W, the full symmetry needs to be preserved even at high temperatures. The distribution density shown in Figure R8 agrees well with the description of the finite temperature string method that the finite-temperature trajectories can be described as a tube in configuration space. The tube center gives the reaction coordinate and results from the average over all trajectories.

We agree that for complex molecular systems at high temperatures, as shown in [The Journal of Chemical Physics 126, 054103 (2007)], the migration pathways may be different since there are no symmetry restrictions. In such a case, the presented TSTI based on a 0 K NEB could still be applied and would provide a first order perturbation result. An extension of the TSTI is also possible, by reconstructing a high-temperature migration path from the mean pathway of many high temperature MD runs (which can be performed on the MTP level). The TSTI can be then calculated around this high temperature mean pathway utilizing additionally a high-temperature effective quasiharmonic reference. Such a development is planned for the future, but it is well beyond our present scope.

Changes:

- In the revised manuscript:

Change #1: Page 3–4: a new discussion has been added: "[...] The reference transition-state configuration, $\{\mathbf{R}_7^{\text{sp}}\}$, is obtained from the climbing-image nudged elastic band³⁸ method performed at zero Kelvin. At high temperatures, even close to the melting point, the MD migration trajectories suggest that, for symmetrical crystalline structures like BCC, the migration pathway remains the same (see Figure S6 in the Supplementary Material and the related discussion). For complex molecular systems³⁹, migration pathways may be modified at high temperatures due to less symmetry restrictions, that would require a special analysis which is out of the scope of the present paper. [...]"

Change #2: Two new references [38] [The J. Chem. Phys. 113, 9901–9904] and [39] [J. Chem. Phys. 126, 054103 (2007)] have been added.

Figure R8: Statistical plot of 754 migration pathways for BCC W collected from MTP-MD at 3400 K.

- In the revised Supplementary Material:

Change #1: A new section “VII MIGRATION PATHWAY AT HIGH TEMPERATURES” has been added based on the reply above.

Change #2: A new Figure S6 has been added.

Comment 3: The authors present their methodology as a significant advancement in the field. However, the manuscript’s description of this methodology leaves substantial uncertainties about its validity. There is a notable lack of comparison with established methods from the literature, such as the string method referenced in paper 32, especially for simpler cases. Given these critical issues, I am unable to endorse the publication of this paper. The unaddressed biases and methodological oversights significantly undermine the credibility and relevance of the findings. Therefore, I must recommend the rejection of this manuscript.

Reply: We have clarified the absence or at least minimal impact of the two possible bias sources as mentioned by you in the previous Comments 1 and 2. In particular, the statistics of the migration pathway derived from additional MD runs shown in Figure R8 agrees well with the description of the string method in Ref. 32 [Phys. Rev. B 66, 052301], suggesting the validity of our method. We hope these revisions and explanations have addressed your concerns.

Minor points

Comment 4.1: For all equations involving F^{ah} , the quasi-harmonic free energy is missing (e.g., equation 2 and equation 16).

Reply: Equations (2) and (16) are specific expressions for describing the anharmonic free energy contribution instead of the full free energy. Therefore, we do not need to formulate the quasiharmonic free energy in these equations.

Comment 4.2: There is a lack of data (such as potential design, fitting database, error analysis, etc.) provided for the used MTP machine learning potential, with the exception of RMSE for forces. For a review as Nature Com, where there is enough space for clarifying methods, supp. materials etc is quite surprising.

Reply: We thank you for this constructive comment. Following your suggestions, in both the revised manuscript and Supplementary Material, we have significantly revised/added computational details and discussions including the data and discussions for the development of the

machine learning potential. In particular, the revised Supplementary Material is significantly extended and re-organized by adding subsections for the machine learning potential design, the fitting procedure and the training dataset. The error analysis of the machine-learning potential has been added in the revised manuscript by new discussion text and new figures. We hope these revisions will make our manuscript more informative.

Changes:

- In the revised manuscript:

Change #1: Page 2: An error analysis of the machine learning potential has been added: “The MTP is trained on a large DFT dataset (2591 structures) including bulk, vacancy, and transition-state configurations sampled from high-temperature MD. The root-mean-square error (RMSE) is only 1.9 meV/atom in energies and 0.15 eV/Å in forces, respectively. Consequently, the MD energies (2591 energies in total) and forces (nearly one million forces in total) predicted by the MTP show a strong correlation with the DFT data, cf. Figure 2. The good performance of the MTP ensures an accurate description of the vibrational phase space for the considered bulk, vacancy, and transition-state configurations, and boosts up the direct upsampling convergence. More details about the MTP are given in the Supplementary Material.”

Change #2: A new Figure 2 showing correlations of the energies and forces predicted by MTP with DFT have been added (repeated here in Figure R9).

- In the revised Supplementary Material:

Change #1: Page 2: A new section C “Development of the machine learning potential” with two sub-sections “Potential form” and “Training database and active learning” has been added where the related computational details are comprehensively discussed.

Change #2: A new Figure S2 has been added.

Figure R9: **Accuracy of the moment tensor potential (MTP).** Correlation plots of the energies (a) and forces (b) of the MTP vs. DFT and the corresponding root-mean-square errors (RMSEs)..

Comment 4.3: Equation (18) presents thermodynamic perturbations between DFT and temperature DFT data on the MTP canonical ensemble. However, there is no data provided about distribution overlaps, even though MTP potentials are typically used in an active learning framework due to their energy landscape instability.

Reply: We agree with you that the configurations sampled in Equation (18) are from the MTP MD trajectory. As clearly observed from the new Figure 2(b) (Figure R9 above) in the

revised manuscript, the MTP forces and DFT forces are highly correlated, suggesting that their vibrational phase spaces exhibit high similarity and thus the free energy perturbation theory can be applied. It has also been shown that the electronic excitation has negligible impact on the vibrational degree of freedom, see [Phys. Rev. B 84, 214107]. This suggests that the vibrational spaces of DFT with and without the electronic excitation are rather similar.

Changes:

- In the revised manuscript:

Change #1: Page 9: A discussion text has been added: “Due to the accurate MTP forces as compared to DFT (Figure 2(b)), the free energy perturbation theory can also be applied to capture the free energy difference between DFT with and without electronic excitations utilizing the MD snapshots generated by the MTP.”

Comment 4.4: The authors claim that their framework can be, or is being, used for High-Entropy Alloys (HEA). However, HEAs are only mentioned briefly in the main manuscript, and only a single result is provided in the supplementary material.

Reply: The results for HCP high entropy alloys (HEAs) shown in the manuscript and in the Supplementary Material verify and substantiate the general applicability of the proposed TSTI approach, i.e., the TSTI approach can be applied not only to simple pure metals like tungsten but also to *complex multicomponent high-entropy alloys* with explicit anharmonicity taken into account for the transition state. As a critical step in the proposed computational framework, the well-behaved thermodynamic integration curve and the corresponding stable molecular dynamics trajectories verify the robust performance and successful application of TSTI in complex materials systems. Nevertheless, a complete simulation study of the diffusion in HEAs with DFT accuracy requires dramatically increasing computational effort due to the fact that the significant chemical complexity comes into play. Statistical sampling in both configurational and vibrational degrees of freedom must be performed and converged to obtain reasonable diffusivities. Such an investigation goes beyond the scope of the present study and is the focus of a separate project currently we are working on.

Changes:

- In the revised manuscript:

Change #1: Page 8: A discussion text has been added: “A complete simulation study of the diffusion in HEAs with DFT accuracy requires an increased computational effort due to the additional chemical complexity. Statistical sampling in both configurational and vibrational degrees of freedom must be performed and converged to obtain reasonable diffusivities. Such an investigation goes beyond the scope of the present study and is the focus of a separate project.”

- In the revised Supplementary Material:

Change #1: A discussion text has been added in “VIII. APPLICATION OF TSTI IN HIGH-ENTROPY ALLOYS”: First TSTI results for a five-component HCP high entropy alloy (HEA), $\text{Al}_{15}\text{Hf}_{25}\text{Sc}_{10}\text{Ti}_{25}\text{Zr}_{25}$, are shown in Figure S7. The HEA was modeled with a special quasirandom structure with 95 atoms ($4 \times 4 \times 3$ expansion of the conventional HCP unit cell; cf. Figure S7(b)). The anharmonic free energy of a Zr atom at the transition state (encircled in the figure) was investigated with TSTI at 1000 K. Specifically, the full λ dependence of $\langle E^{\text{full}} - E^{\text{qh}} \rangle_{\lambda}$ from a quasiharmonic reference to an MTP (RMSE of 2.5 meV/atom) was calculated and is plotted in Figure S7(a). All transition-state trajectories entering the calculation are stable and the resulting curve is well-behaved. These results substantiate the robust performance of the TSTI approach also for chemically complex materials.

Response to Reviewer #4

Comment: I co-reviewed this manuscript with one of the listed reviewers.

Reply: We thank you for your time and effort on the review of the manuscript. We hope that the responses above and the revised manuscript have addressed your concerns.

July 16, 2024

Dear Reviewers,

We thank all four of you for the reports and the constructive comments. Following, in particular, the comments from Reviewer #3, we have further validated our computational procedure and revised the manuscript accordingly with the additional calculations and results. Below is a detailed response to each comment and the resulting revisions. The main changes are highlighted in magenta in the revised manuscript, the Supplementary Material, and this letter. We hope that the revised manuscript will be suitable for publication in Nature Communications.

With best regards,
Xi Zhang, on behalf of all co-authors

Responses to Reviewer #1

Remarks to the Author: The authors have successfully addressed my concerns in the response letter and revised manuscript. The more detailed discussions on the contribution of electronic excitation is more clear and the outlook on how to generalize this method in unstable structure is insightful. The atomic trajectory is helpful for visualization as well. I would recommend this manuscript for publication.

Reply: We thank you for your appreciation of our revision of the manuscript and the recommendation for publication.

Remarks on code availability: If possible, the authors are suggested to provide input file samples.

Reply: We will provide input file examples together with the resubmission of the manuscript.

Response to Reviewer #2

Remarks to the Author: The authors have address my concerns to my satisfaction.

Reply: Thank you for your feedback and for your time to review our manuscript. We are glad to hear that we have addressed your concerns to your satisfaction.

Responses to Reviewer #3

Comment #1: [...] Why the equations R17, R18 and R19 from the answer are correct. [...] Why the present method is incorrect. [...] the method has a mathematically weak foundation. [...] (The full comments are not repeated here.)

Reply: We thank you for the detailed description of your concerns and the corresponding supporting references, i.e., J. Chem. Phys. 115, 9169 (2001) and C. Chipot, A. Pohorille (2007).

After carefully reading your comments and the references, we realized that we need to elucidate the significant distinction between the thermodynamic integration we used and the mean-force-based integration described in the comments.

1. The TSTI approach differs fundamentally from the description in the comments.

Free energy calculations via thermodynamic integration (TI) can be implemented along either one of *two fundamentally distinct ways* (labeled Method #1 and Method #2 below). These two ways are distinguished by the thermodynamic integration pathway (TIP) and the corresponding coupling parameter (CP) that describes the changes of the target system [termed “order parameter” in Chapter 4 of the book “Free Energy Calculations: Theory and Applications in Chemistry and Biology”, C. Chipot, A. Pohorille (2007)]:

- Method #1:

TIP: A physical transformation path (e.g., the migration path of the diffusing atom).

CP: A *reaction coordinate* or *reaction path*.

- Method #2:

TIP: An auxiliary path connecting two potential energy surfaces (e.g., harmonic potential and anharmonic potential).

CP: An auxiliary interpolation (usually linear) variable between the two potentials.

The TI described in your comments corresponds to “Method #1”, for which the dynamics of the coupling parameter (reaction coordinate) can be described by Newton’s equations of motion. The free energy difference between positions r_1 and r_2 along the reaction coordinate (coupling parameter) $\Delta F^{r_1 \rightarrow r_2}$ can be written as

$$\Delta F^{r_1 \rightarrow r_2} = F(r_2) - F(r_1) = \int_{r_1}^{r_2} \frac{\partial F(r)}{\partial r} dr \quad (\text{R1})$$

$$= \int_{r_1}^{r_2} \left\langle \frac{\partial E(r)}{\partial r} \right\rangle_r dr \quad (\text{R2})$$

$$= \int_{r_1}^{r_2} -\langle f \rangle_r dr. \quad (\text{R3})$$

The equality of Eqs. (R1) and (R3) demonstrates that the free energy as a function of the reaction coordinate can be computed as the potential of the mean force $\langle f \rangle$. To sample or calculate the average in Eq. (R2), both constrained (fixed r with external force) and unconstrained dynamics (atoms can move freely along the reaction coordinate) can be applied.

The TI/TSTI used in the present work, in contrast, corresponds to “Method #2”. The calculated free energy corresponds to a certain *atomic configuration, either the equilibrium configuration or the saddle point configuration*. The free energy difference between the two potentials reflects the difference between harmonic vibrations (“pot1”, described by the dynamical matrix) and the full (anharmonic) vibrations (“pot2”, described by the moment tensor potential (MTP)). The thermodynamic integration pathway is designed with an auxiliary coupling parameter $\lambda \in [0, 1]$ such that, for $\lambda = 0$, the potential of the system $E(\lambda)$ driving the atomic motion corresponds to “pot1” while for $\lambda = 1$, the potential of the system corresponds to “pot2” and a linear mixing of pot1 and pot2 when $0 < \lambda < 1$. The potential function of the system as a function of λ is written as,

$$\begin{aligned} E(\lambda) &= (1 - \lambda)E^{\text{pot1}} + \lambda E^{\text{pot2}} \\ &= E^{\text{pot1}} + \lambda(E^{\text{pot2}} - E^{\text{pot1}}). \end{aligned} \quad (\text{R4})$$

In this case, it is clear that the coupling parameter λ is not the reaction coordinate (e.g., r in Method #1) and no equations of motion exist naturally for the coupling parameter λ . The corresponding forces

$$\mathcal{F}(\lambda) = (1 - \lambda)\mathcal{F}^{\text{pot1}} + \lambda\mathcal{F}^{\text{pot2}} \quad (\text{R5})$$

are not representing any mean force $\langle f \rangle$ but simply the actual forces acting on the atoms. The free energy difference between “pot2” and “pot1” can be written as

$$\begin{aligned}
\Delta F^{\text{pot1} \rightarrow \text{pot2}} &= F(\lambda = 1) - F(\lambda = 0) = \int_{\lambda=0}^{\lambda=1} \frac{\partial F(\lambda)}{\partial \lambda} d\lambda \\
&= \int_{\lambda=0}^{\lambda=1} \left\langle \frac{\partial E(\lambda)}{\partial \lambda} \right\rangle_{\lambda} d\lambda \\
&= \int_{\lambda=0}^{\lambda=1} \langle E^{\text{pot2}} - E^{\text{pot1}} \rangle_{\lambda} d\lambda,
\end{aligned} \tag{R6}$$

where the last step in Eq. (R6) is obtained from the derivative of Eq. (R4) with respect to λ . Comparing Eq. (R6) with Eqs. (R1)–(R3), it is evident that the free energy difference ΔF and the coupling parameter described in “Method #1” and “Method #2” are fundamentally different.

2. TSTI approach is not constrained dynamics.

Practically, the evaluation of the integral in Eq. (R6) is performed via a set of equilibrium *NVT* MD simulations at different but *fixed* λ values. MD simulations at a fixed coupling parameter λ do not, in any way, require constrained dynamics. The potential and the forces are strictly defined via Eq. (R4) and (R5). This fundamentally differs from “Method #1” where fixing the coupling parameter r (reaction coordinate) requires constrained dynamics with explicit external forces.

In the TSTI approach for the saddle point configuration, all atoms can move freely in all vibrational degrees of freedom. The only modification is that, for a *single unstable mode*, the unstable forces in both “pot1” and “pot2” are replaced with the *same, stable* harmonic forces along full the thermodynamic integration path (from $\lambda = 0$ to $\lambda = 1$) to circumvent the instability. The vibrational behavior of this single unstable mode becomes then the *same and stable* for both “pot1” and “pot2” along the full thermodynamic integration path. When the potential energy difference in Eq (R6), i.e., $E^{\text{pot2}} - E^{\text{pot1}}$, is computed, there is no explicit contribution from this single unstable mode to the free energy difference $\Delta F^{\text{pot1} \rightarrow \text{pot2}}$ due to the cancellation. Free energy contributions from all other vibrational degrees of freedom are fully taken into account. This is consistent with the harmonic transition state theory in which the free energy contribution from the single unstable mode is excluded.

3. Calculation of migration free energy.

By applying the TI/TSTI approach, we compute the free energy difference between the quasiharmonic potential and the full anharmonic potential (MTP), i.e., anharmonic free energy ΔF^{ah} , for the equilibrium vacancy configuration ($\Delta F_{\text{vac}}^{\text{ah}}$) and for the transition state saddle point configuration ($\Delta F_{\text{TS}}^{\text{ah}}$), separately. Combing all different free energy contributions, the full free energy for both types of configurations can be written as

$$F_{\text{vac}} = E_{\text{vac}}^{\text{0K}} + \Delta F_{\text{vac}}^{\text{qh}} + \Delta F_{\text{vac}}^{\text{ah}} + \Delta F_{\text{vac}}^{\text{el}} \tag{R7}$$

$$F_{\text{TS}} = E_{\text{TS}}^{\text{0K}} + \Delta F_{\text{TS}}^{\text{qh}} + \Delta F_{\text{TS}}^{\text{ah}} + \Delta F_{\text{TS}}^{\text{el}}. \tag{R8}$$

The vacancy migration free energy is then calculated as

$$F_{\text{mig}} = F_{\text{TS}} - F_{\text{vac}}. \tag{R9}$$

Changes:

- In the revised manuscript:

Change #1: Based on the explicit validation of our computed data against brute-force MD simulations (see Comment 2), we were able to detect a mistake in the technical implementation of the TSTI method. This mistake led to an inconsistent treatment of the

MTP potential energies $E^{\text{full}}(\{\mathbf{R}_I\})$ entering the thermodynamic integrand in Eq. (3) of the manuscript (repeated here):

$$\Delta F_{\text{MTP}}^{\text{qh} \rightarrow \text{full}} = \int_0^1 d\lambda \langle E^{\text{full}}(\{\mathbf{R}_I\}) - E^{\text{qh}}(\{\mathbf{R}_I\}; \underline{\underline{\mathcal{D}_s}}) \rangle_\lambda, \quad (\text{R10})$$

as compared to the forces. Specifically, when the stabilized MTP forces are applied to drive the atomic motion along the unstable mode, the corresponding MTP potential energy for computing the integration (Eq. R12) needs to be modified/updated consistently as

$$E^{\text{full}} = E^{\text{full}}(\{\mathbf{R}_I\}) + [E^{\text{qh}}(\{\mathbf{R}_I^{\text{sp}} + \mathbf{u}'_I\}; \underline{\underline{\mathcal{D}_s}}) - E^{\text{full}}(\{\mathbf{R}_I^{\text{sp}} + \mathbf{u}'_I\})]. \quad (\text{R11})$$

We have corrected the implementation, redone all saddle point calculations, and updated the corresponding data and figures. The corresponding theoretical description on Page 2 has been modified as: “[...] Specifically,

$$\Delta F_{\text{MTP}}^{\text{qh} \rightarrow \text{full}} = \int_0^1 d\lambda \langle E_s^{\text{full}}(\{\mathbf{R}_I\}) - E^{\text{qh}}(\{\mathbf{R}_I\}; \underline{\underline{\mathcal{D}_s}}) \rangle_\lambda \quad (\text{R12})$$

represents a thermodynamic integration along the coupling parameter $\lambda \in [0, 1]$, with the thermodynamic average $\langle \dots \rangle_\lambda$ of the difference between the quasiharmonic energy E^{qh} and the *stabilized* full vibrational MTP energy E_s^{full} (discussed below) [...]”. On page 3, the following equation and text have been added: “Consistently, the corresponding MTP potential energy entering Eq. (3) is computed as

$$E_s^{\text{full}}(\{\mathbf{R}_I\}) = E^{\text{full}}(\{\mathbf{R}_I\}) + [E^{\text{qh}}(\{\mathbf{R}_I^{\text{sp}} + \mathbf{u}'_I\}; \underline{\underline{\mathcal{D}_s}}) - E^{\text{full}}(\{\mathbf{R}_I^{\text{sp}} + \mathbf{u}'_I\})], \quad (\text{R13})$$

where E^{full} is the original unstabilized MTP potential energy.”

The main update concerns Figure 3(b) and Figure 6 (repeated below as Figure R1 and Figure R2). For the calculated migration Gibbs energies in Figure R1, while the “qh + ah” curve and the “qh + ah + el” curve show an overall upward shift, the general non-linear curvatures as well as the general conclusions in particular regarding the high-temperature regime (gray shaded area) remain the same. The calculated temperature-dependent diffusivities in Figure R2(a) show a better agreement with experiment, i.e., our computed data lies very close to the experimental data. The slope in Figure R2(b) remains almost unchanged. The corresponding discussion text on Page 4 has been modified in the revised manuscript as: “For both $G_{\text{form}}(T)$ and $G_{\text{mig}}(T)$, a strong and clearly *non-linear* temperature dependence is observed, particularly in the temperature range where diffusion is measured ($T > 0.46 T^{\text{melt}}$, gray shaded area).”

- In the Supplementary Material:

Change #1: Results computed using the TSTI approach, i.e., Figures S3, S5, and S9(a) have been updated.

Comment #2: [...] The results are biased. [...]

Reply: In order to quantitatively check the accuracy of our TSTI approach and verify the quality of the chosen saddle point configuration \mathbf{R}_I^{sp} and unstable mode \mathbf{w}_{imag} at 0 K as the reference, we additionally performed direct “brute-force” molecular dynamics (MD) simulations using the MTP to extract diffusivities at three high temperatures, i.e., 3000 K, 3200 K, and 3400 K. As the diffusivity computed from the direct MTP MD simulations eliminates any 0 K based assumptions, it can be considered as a theoretical benchmark at the MTP level.

An example of the mean-squared displacement as a function of time is plotted in Figure R3 and the resulting diffusivities are shown in Figure R4 and compared with the MTP TSTI results. It is clear that diffusivities computed from both methods agree very well not only in the absolute

Figure R1: Migration Gibbs energies from the proposed TSTI approach.

Figure R2: **Non-Arrhenius self-diffusion of tungsten.** (a) Arrhenius plots of self-diffusivity in BCC tungsten calculated with the proposed *ab initio* machine-learning TSTI approach with all finite-temperature excitations taken into account (red circles and lines) on the homologous temperature scale in comparison to experimental data from Mundy et al.¹², Andelin et al.⁶⁴, Pawel et al.⁶⁵, and Arkhipova et al.⁶⁶. (b) A comparison of the temperature-dependent slopes of the Arrhenius plots in (a).

values but also in the temperature dependence. This demonstrates that employing the 0 K \mathbf{R}_I^{SP} and \mathbf{w}_{imag} introduces negligible bias or inaccuracy for high temperature diffusivity calculations. In fact, this can be well understood from Figure S6 of the Supplementary Material that, in particular for a symmetrical crystalline structure, the high-temperature migration pathways follow the 0 K NEB pathway within an ensemble average.

Changes:

- In the Supplementary Material:
 - Change #1: The computational details and key results of the direct MTP MD simulations have been added in the Supplementary Material as a new section “VIII. DIRECT MTP MD SIMULATIONS”
 - Change #2: Two new figures, Figures S7 and S8, have been added.

Figure R3: Mean squared displacement as a function of time at 3400 K. Gray lines are the results of each independent MD run (50 in total) and the black line shows the mean values.

Figure R4: Arrhenius plot of self-diffusivity computed from the present TSTI approach and direct MD simulations using MTP.

Response to Reviewer #4

Remarks to the Author: I co-reviewed this manuscript with one of the listed reviewers.

Reply: We thank you for your time and effort on the review of the manuscript.

October 27, 2024

Dear Reviewers,

We thank Reviewer #2 for the additional effort on the re-assessment of our work and the very positive feedback. We also thank Reviewers #3 and #4 for explaining their concerns with a 2D model. Below is a detailed response to each comment and the resulting revisions. The main changes are highlighted in magenta in the revised manuscript and this letter. We hope that the revised manuscript will be suitable for publication in Nature Communications.

With best regards,
 Xi Zhang, on behalf of all co-authors

Response to Reviewer #2

Remarks to the Author: I have looked carefully at the reviewers' objections to the present manuscript as well as the authors' response. I believe that the underlying reasons for the objections result from a misunderstanding of the thermodynamic integration procedure followed by the authors. The authors are using rigorous transition state theory to integrate out the unstable mode according to practice established over a long time, modified by their own state-of-the-art methods. It is thus my opinion that the authors are correct and therefore I do not think there are further issues with the present manuscript.

Reply: We thank you for your careful check of our TSTI approach and your confirmation of its correctness.

Response to Reviewer #3/#4

Remarks to the Author: It seems that the authors remain steadfast in their positions. Please find the report in the attached pdf file. There, you will also find clear proof of the incorrectness of their method. Consequently, we have reached the same conclusion: while the present paper seems interesting, the method presented, in its current form, is simply incorrect. Although some aspects could potentially be improved or corrected, none of these improvements were evident in the authors' reply. Therefore, with regret, I must say that in its present form, I can only recommend rejection.

Reply: We thank you for again reconsidering your response and further explaining your concerns with a 2D model. The conclusion from your Eq. (5),

$$\mathcal{F}_s(x, y) = -[2x + \partial_x f(x, y) - \partial_x f(0, y)]\mathbf{e}_x - [2\alpha y + \partial_y f(x, y) - \partial_y f(0, y)]\mathbf{e}_y, \quad (\text{R1})$$

is not correct. The red terms do not bias the sampling in any way. These terms are rather responsible for removing a significant part of the unwanted coupling between the one stabilized mode and the $3N - 1$ stable modes. The remaining influence beyond what is removed by the red terms can be analyzed and quantified by varying α in our TSTI approach. To understand the impact of α , it is not sufficient to consider only energies and forces as you did in your 2D model. It is also necessary to include the concept of canonical sampling. Variation of α modifies

the canonical probability distribution primarily in the vicinity of the stabilized mode. In the original manuscript, we had already tested three different α values. Now, we have substantially extended the analyzed α range. The conclusion remains the same, i.e., that little impact of α is seen in the final results.

Figure R1 shows the change in the anharmonic free energy of the transition state (red circles) as a function of α (top x -axis), expressed on the bottom x -axis in terms of the resulting stabilized frequency. The representation in frequency has the benefit of putting the α values in relation to the native frequency spectrum of the transition state. The latter is represented by the phonon density of states (shown by the gray curve and shading). For α values up to a frequency of about 30 meV, the anharmonic free energy increases and then saturates, i.e., the α dependence reaches a plateau above the highest frequency in the system. It is important to note that the observed changes in F^{ah} are comparatively small ($\approx 1\%$) and close to the statistical resolution limit of the calculations, even when using a machine-learning potential. To put these changes into perspective, we have calculated the impact on the final diffusion coefficient. For example, changing F^{ah} by 0.03 eV/vacancy introduces a change of the diffusion coefficient (cm^2/s) from 5.411×10^{-9} to 4.819×10^{-9} at a representative temperature $T = 3000$ K. This corresponds to typical uncertainties of very accurate experimental diffusion tracer measurements when the independent measurements at nominally the same temperature T are compared.

Although the impact of α is rather small, to ensure fully converged anharmonic free energies and to pursue the highest accuracy of the calculations, we have updated the Gibbs energy of migration and the diffusivities using $\alpha = 30$ in the revised manuscript. The diffusivities show better agreement between DFT and experiment (Fig. 6 in the revised manuscript), as well as between TSTI and direct MD simulations (Fig. S8 in the Supplementary Material).

Figure R1: Variation of the anharmonic free energy (red open circles) with different stabilized harmonic vibrational frequencies for the single unstable mode. The corresponding α values are marked on the opposite side of the horizontal axis. The characteristic phonon density of states (DOS) for the $3N-1$ stable modes at the transition state is shown with gray shading.

Based on the current findings, the future applicants of the TSTI approach are advised to analyze carefully the phonon density of states and chose the parameter α accordingly.

Changes:

- In the revised manuscript:

Change #1: Figs. 3(b), 5, and 6 are updated.

Change #2: Page 2: A discussion text has been added: “[...] We set α to be 30, a value ensuring that the resulting stabilized frequency of the unstable mode, $\sqrt{\alpha|\omega_{\text{imag}}^2|}$, is higher than the highest stable frequency of the frequency spectrum of the transition state.

More details and tests on the stabilization parameter α are provided in the Supplementary Material.

Change #3: Page 4: A discussion text has been added: “[...] The indirect impact of the stabilization via the coupling to the stable modes is small and can be controlled with the α parameter (see Supplementary Material).”

- In the revised Supplementary Material:

Change #1: Section II “STABILIZATION PARAMETER IN TSTI” has been revised, including the above discussion.

Change #2: Fig. R1 in this letter has been added as Fig. S3.

**Comments on "Ab initio machine-learning unveils strong anharmonicity in
non-Arrhenius self-diffusion of tungsten"**

I will try to re-write in more comprehensive way the equation formulated by authors. I will drop the index I for atoms as long as there is no specification for a particular atom.

- The ensemble of atomic coordinates is denoted by a $3N$ vector \mathbf{R} in \mathbb{R}^{3N} .
- For the zero K atomic positions at the saddle we also replace \mathbf{R}_I^{sp} by a $\mathbf{R}^{sp} \in \mathbb{R}^{3N}$.

A. Why the equations R17, R18 and R19 from the answer are correct

The mix potential qh-MTP $E^{\text{qh-MTP}}(\lambda, \mathbf{R})$ and the mean force along the reaction coordinate λ i.e. $\mathcal{F}_\lambda(\mathbf{R})$ (the equation R17 and R18) become in my notations as:

$$E^{\text{qh-MTP}}(\lambda, \mathbf{R}) = (1 - \lambda)E^{\text{qh}}(\mathbf{R}) + \lambda E^{\text{MTP}}(\mathbf{R}) \quad (1)$$

$$\mathcal{F}_\lambda^{\text{qh-MTP}}(\mathbf{R}) = -\nabla_{\mathbf{R}} E^{\text{qh-MTP}}(\lambda, \mathbf{R}) = -(1 - \lambda)\nabla_{\mathbf{R}} E^{\text{qh}}(\mathbf{R}) - \lambda \nabla_{\mathbf{R}} E^{\text{MTP}}(\mathbf{R}) \quad (2)$$

The mean force drive the Langevin sampling for a given λ . The ONLY reason for which we can write that the anharmonic free energy as in R19 is because the mean force is the gradient of the mix energy (see above Eq.2). This is true for any minimum. For the saddle points the reality is more nuanced and is detailed in the next section.

B. Why the present method is incorrect

At the saddle point, the authors use a surrogate force which is not the gradient of the energy at the saddle point. Consequently, the free energy derived is incorrect. Moreover, there are implicit and explicit dependencies that are not included in the model. I will present this in a clearer way.

From EQ7 of the paper the used mean force at the saddle point is:

$$\mathcal{F}_\lambda^{\text{qh-MTP}} = (1 - \lambda)\mathcal{F}_s^{\text{qh}}(\mathbf{R}) + \lambda \mathcal{F}_s^{\text{MTP}}(\mathbf{R}) \quad (3)$$

$$= -(1 - \lambda)\nabla_{\mathbf{R}} E_s^{\text{qh}}(\mathbf{R}, \mathbf{D}_s) + \lambda \mathcal{F}_s^{\text{MTP}}(\mathbf{R}) \quad (4)$$

$$\mathcal{F}_s^{\text{MTP}}(\mathbf{R}; \mathbf{w}_{\text{imag}}, \mathbf{R}^{sp}) = \mathcal{F}^{\text{MTP}}(\mathbf{R}) + \left[\mathcal{F}_s^{\text{qh}}(\Pi(\mathbf{R} - \mathbf{R}^{sp})) - \mathcal{F}^{\text{MTP}}(\Pi(\mathbf{R} - \mathbf{R}^{sp})) \right] \quad (5)$$

$$\Pi(\mathbf{R} - \mathbf{R}^{sp}) = \mathbf{R}^{sp} + (\mathbf{w}_{\text{imag}}^\top (\mathbf{R} - \mathbf{R}^{sp})) \mathbf{w}_{\text{imag}} \quad (6)$$

Differently from the paper we denote by Π the projector of the the atomic coordinates along the unstable reaction mode at the saddle point that implicitly depends of the direction of the unstable mode \mathbf{w}_{imag} and the saddle point position at the 0 K \mathbf{R}^{sp} (as the author designed this method).

And 2) the mean force $_s$ of the MTP will depend implicitly and explicitly on those two ZERO K quantities \mathbf{w}_{imag} and \mathbf{R}^{sp} and for these reasons we denote by $\mathcal{F}_s^{\text{MTP}}(\mathbf{R}; \mathbf{w}_{\text{imag}}, \mathbf{R}^{\text{sp}})$.

The reasons for which the present method is questionable:

1. **the method has a mathematically weak foundation.** In Eq. 5 the two mean forces $\mathcal{F}^{\text{MTP}}(\mathbf{R})$ and $\mathcal{F}_s^{\text{qh}}(\Pi(\mathbf{R} - \mathbf{R}^{\text{st}}))$ are the gradients of two well identified energies $E^{\text{MTP}}(\mathbf{R})$ and $E^{\text{qh}}(\mathbf{R}, \mathbf{D}_s)$. The reason for which $\mathcal{F}_s^{\text{qh}}(\Pi(\mathbf{R} - \mathbf{R}^{\text{st}}))$ is the well defined gradient of the $E^{\text{qh}}(\mathbf{R}, \mathbf{D}_s)$ is because its corresponding dynamics is harmonic and all the modes are decoupled. Consequently, the projection is well identified when applied to the harmonic dynamics driven by the normal modes. This is NOT the case for $\mathcal{F}^{\text{MTP}}(\Pi(\mathbf{R} - \mathbf{R}^{\text{sp}}))$. The constrained dynamics do not ensure that the mean force is the gradient of the constrained dynamics. For extensive presentation I recommend to the authors to look in standard text for the free energy sampling under constraints such as seminal paper, "Calculating free energies using average force" E. Darve, A. Pohorille, J. Chem. Phys. 115, 9169 (2001) or some classic books on this topic such as "Free Energy Calculations: Theory and Applications in Chemistry and Biology" C. Chipot, A. Pohorille (2007). The publication, which is presented by the authors J. Chem. Phys. 126, 054103 (2007) is an example of how the constraint dynamics is treated correctly. Therefore, it is not the case of the present approach.

Consequently, the reconstruction of the free energy from the questionable mean force in Eq. 5 is flawed. Given the lack of other details from the authors except for the general equations R17-R19, I conclude that the method is mathematically incomplete and irrelevant for the community.

2. **the results are biased** Let's suppose that we find a correct way to extract free energy $F^{\text{ah}}(\mathbf{w}_{\text{imag}}, \mathbf{R}^{\text{sp}})$ from the mean force at the saddle point $\mathcal{F}_s^{\text{MTP}}(\mathbf{R}; \mathbf{w}_{\text{imag}}, \mathbf{R}^{\text{sp}})$ (and eventually from $\mathcal{F}_\lambda^{\text{qh-MTP}}(\mathbf{R}; \mathbf{w}_{\text{imag}}, \mathbf{R}^{\text{sp}})$). The corresponding free energy will intrinsically depend on ZERO K quantities such as \mathbf{w}_{imag} and \mathbf{R}^{sp} . The entire literature presents examples for which these quantities are not defined at finite temperatures. Some methods propose a variational determination of these two quantities. However, the present method does not propose such a sanity check. Consequently, the results can only be biased. The bias can be either small or large. However, there is no proof in any direction. The authors presented a kind of 'visual' analysis of the saddle point position. The results presented are pictorial not quantitative and cannot justify the susceptibility of the method to avoid bias.

Both points described above are crucial for treating the physics of high-temperature constrained

dynamics. Neither of them was treated correctly. Accurate analysis and benchmarking of the method are missing. The direct comparison of DFT versus experiment is irrelevant because compensation errors can occur. One of the points above, concerning the incompleteness of the present method (specifically the last point, point 2), is recognized even by the authors, with the justification: 'We agree that... at high temperatures, the migration pathways may be different... An extension of the TSTI is also possible... Such a development is planned for the future.' In my opinion, this iterative and incremental approach can lead to false physical results and conclusions. These risky points, in my humble opinion, are not acceptable for a prestigious journal such as Nature Communications.

My conclusion is the same as after the first report. An attentive reader must identify that I am saying exactly the same things, but now with equations instead of words. The present paper seems to be interesting, but the presented method, in this form, is simply incorrect. Probably some aspects can be improved, some of them corrected. However, none of those signs of improvement were seen in the author's reply. Therefore, with regret, I must say that in its present form, I can only recommend rejection.

Comments on "Ab initio machine-learning unveils strong anharmonicity in
non-Arrhenius self-diffusion of tungsten"

In the new version of the article / response letter, the authors clarify certain aspects of the methodology developed. These aspects are essential because the scientific credibility of the paper depends on them. The synthesis of the answer and the claim of the authors lie in this postulated reality: "a modified convex sampling at the saddle point can be used instead of constraint sampling." The authors not provide any foundation of this statement; it is merely a belief. Nevertheless, the present methodology is mathematically ill defined. In order to sample the free energy of a transition state, the canonical probability measure can only be defined on a manifold of phase space ["Free energy computations: a mathematical perspective" T.Lelievre, M. Rousset and G. Stoltz, Imperial College Press,2010]. Furthermore, it cannot be considered an approximation or model, as it lacks any physical foundation.

Under this postulate the authors compare their results for the self-diffusion coefficient from equation (11) of the paper with extensive MD trajectories, where the self-diffusion can be directly estimated by the mean square displacement. Comparison between their method and long MD trajectories is presented in Figure S8 (supplementary material). By some compensation of errors, the results are in relatively good agreement. However, the authors do not directly compare activation free energies from the both calculations. Moreover, the authors do not provide any quantitative criterion to verify the validity of their results for a given system. The only validity argument presented in the supplementary material is the "bcc symmetries" of migration pathways at high temperatures, which is not quantitatively admissible.

To be brief and to the point: here, we reveal the weak mathematical foundation of the present method in the framework of an easy-to-understand 2D toy model. Let's define the energy of the system as a 2D function at any given point: $\mathbf{r} = xe_x + ye_y$, with $x, y \in \mathbb{R}$, $\mathbf{r} \in \mathbb{R}^2$ and e_{xy} are the normalized Cartesian axis. The energy reads:

$$E(x, y) = x^2 - y^2 + f(x, y), \quad (1)$$

where x^2 are the $3N - 1$ stable coordinates at the saddle point (in the paper notations) and $-y^2$ is the unstable mode. Evidently, without loss of generality, the saddle point coordinate is located at $(0, 0)$. The function $f(x, y)$ is a coupling function that ensure the anharmonic part of the potential. For this toy-model the $\hat{\mathbf{w}}_{\text{img}}$ vector described in the paper correspond to e_y . From the definitions

and equations described in the paper:

$$\mathcal{F}(x, y) = -\nabla E(x, y) = -(2x + \partial_x f(x, y))\mathbf{e}_x + (2y - \partial_y f(x, y))\mathbf{e}_y, \quad (2)$$

$$\mathcal{F}_s^{\text{qh}}(0, y) = -2\alpha y\mathbf{e}_y, \quad (3)$$

$$\mathcal{F}(0, y) = -\nabla E(x', y')|_{x'=0, y'=y} = -\partial_x f(0, y)\mathbf{e}_x + (2y - \partial_y f(0, y))\mathbf{e}_y. \quad (4)$$

We denote by $\partial_k f(0, y)$ the following expression $\partial_k f(x', y')|_{x'=0, y'=y}$. The resulting forces used for the sampling gives:

$$\begin{aligned} \mathcal{F}_s(x, y) &= \mathcal{F}(x, y) + [\mathcal{F}_s^{\text{qh}}(0, y) - \mathcal{F}(0, y)] \\ &= -[2x + \partial_x f(x, y) - \partial_x f(0, y)]\mathbf{e}_x - [2\alpha y + \partial_y f(x, y) - \partial_y f(0, y)]\mathbf{e}_y \end{aligned} \quad (5)$$

From these equations is more than obvious that the sampling is biased by $\partial_x f(0, y)$, $\partial_y f(0, y)$ along \mathbf{e}_x and \mathbf{e}_y , respectively. It is easy to choose anharmonic coupling function where the canonical sampling will fail.

This is a direct consequence of the two postulates that are at the foundation of this study (a) *TSTI approach is not a constrained dynamics* and (b) *a modified convex sampling at the saddle point can be used instead of a constraint sampling*. Consequently, the method can not ensure a convex sampling for TI at saddle point. This is in opposite with the claim of authors: their method is not wrong and give quantitative results for specific case. Nevertheless, the mathematical reality is cruel: this methodology is ill defined even in 2D case. The general case is obvious that has no any proof of mathematical foundation, robustness or convergence. In front of those signs of incorrectness we let to the editors the responsibility to choose if this "specific" methodology should be published in Nature Communication.